# PRICE: direct and robust detection of microRNAs at single-nucleotide resolution

Buhua Wang [1,20], Shuai Zhou[2,20], Xi Zhang[1], Rui Wang[3,4,5,6], Shuo Huang[7], Anyi Li[1], Haowei He[8], Yingai Zhang[8], Yangdao Wei[9], Zhiqing Yang[1], Fengge Song [1], Xiangpeng Li[10], Xinyi Wan[11], Yi Shen[1], Chunxin Ma[1], Haimei Mao[12], Rodrigo Ledesma-Amaro [13,14,15], Dapeng Yin[16], Qingshan Wei [17], Ruijie Deng [18], Tao Yang[7], Shanrong Liu[19], Yi Wan [1] ✉ & Chuanbin Mao [7] ✉

Accurate single-nucleotide discrimination of miRNA is clinically vital because small sequence variations can have significant phenotypic and clinical consequences, yet existing techniques can only detect single nucleotide variations (SNVs) at specific loci. Here, we present a generalized peptide nucleic acid (PNA) mediated CRISPR/Cas13a system (PRICE), enabling detection of SNVs in miRNA sequence without sacrificing the sensitivity. PRICE utilizes PNA blockers fully complementary to non-target miRNAs (e.g., miRNAs containing SNVs at loci of no interest) but not to the target miRNA. These blockers selectively hybridize with and inhibit non-target sequences in samples (serum, cells, or tissues). Only the unhybridized target miRNA then binds to crRNA within the Cas13a complex, activating Cas13a to cleave a fluorescent reporter-quencher linker, generating a detectable signal (~10 fM limit). By designing a panel of PNAs against SNVs, PRICE provides a versatile, amplification-free platform for precise miRNA analysis, advancing cancer diagnosis, prognosis, and biology.

microRNAs (miRNAs) play a pivotal role in post-transcriptional gene expression regulation, and minor variations in miRNA sequences can have far-reaching phenotypic and clinical consequences. For instance, a single nucleotide polymorphism in the let-7 miRNA within the Kirsten rat sarcoma viral oncogene homologue (KRAS) untranslated region has been linked to colorectal cancer[1]. The presence of single-nucleotide variations (SNVs) within miRNAs can disrupt their normal function, leading to cellular dysregulation and potentially contributing

[1]School of Marine Sciences (State Key Laboratory of Marine Resource Utilization in South China Sea), Hainan University, Haikou, China. [2]Department of Hepatobiliary Surgery, Haikou Affiliated Hospital of Central South University Xiangya School of Medicine, Haikou, China. [3]NHC Key Laboratory of Tropical Disease Control, Hainan Medical University, Haikou, China. [4]Engineering Research Center for Hainan Bio-Smart Materials and Bio-Medical Devices, Hainan Medical University, Haikou, China. [5]Key Laboratory of Hainan Functional Materials and Molecular Imaging, Hainan Medical University, Haikou, China. [6]School of Life Sciences and Medical Technology, Hainan Medical University, Haikou, China. [7]Department of Biomedical Engineering, The Chinese University of Hong Kong, Shatin, Hong Kong SAR, China. [8]Central Laboratory, Affiliated Haikou Hospital of Xiangya Medical College, Central South University, Haikou City Key Laboratory of Clinical Medicine, Haikou, China. [9]Department of Clinical Hematology and Institute of Materia Medica, College of Pharmacy and Laboratory Medicine Science, Army Medical University, Chongqing, China. [10]Department of Chemistry and Biochemistry, Florida State University, Tallahassee, FL, USA. [11]School of Environmental Science and Engineering, Hainan University, Haikou, China. [12]Products Quality Supervision and Testing Institute of Hainan Province, Haikou, China. [13]Department of Bioengineering and Imperial College Centre for Synthetic Biology, Imperial College London, London, UK. [14]Bezos Centre for Sustainable Protein, Imperial College London, London, UK. [15]UKRI Engineering Biology Mission Hub on Microbial Food, Imperial College London, London, UK. [16]Hainan Center for Disease Control and Prevention, Haikou, China. [17]Department of Chemical and Biomolecular Engineering, Emerging Plant Disease and Global Food Security Cluster, North Carolina State University, Raleigh, NC, USA. [18]College of Biomass Science and Engineering, Healthy Food Evaluation Research Center, Sichuan University, Chengdu, China. [19]Department of Laboratory Diagnostics, Changhai Hospital, Navy Military Medical University, Shanghai, China. [20]These authors contributed equally: Buhua Wang, Shuai Zhou. ✉e-mail: 993602@hainanu.edu.cn; cmao@cuhk.edu.hk

to the development of various diseases[2–4], such as breast cancer[5] and gastric cancer[6]. Thus, discriminating SNVs in miRNAs is crucial for understanding their functional implications[7], disease associations[8,9], therapeutic opportunities[10,11], and biomarker discovery[12–14], ultimately advancing our knowledge and potential research in the field of miRNA.

The gold standard methods to identify and profile miRNAs rely on the reverse transcription quantitative polymerase chain reaction (RT-qPCR)[15]. Different from long RNA targets, the discrimination of SNVs in miRNAs using RT-qPCR relies on the specific primers design to extend the target template, capturing either the wild-type or variant allele during reverse transcription. However, this method has a key limitation, since it easily suffers biases from the variance of GC content in miRNA, which leads to a wide variation in melting temperatures ($T_m$) for annealing reactions[16,17]. RNA sequencing (RNA-seq)[18,19] is another powerful technology to identify SNVs, but the limited miRNA abundance and the short length of miRNA (~22 nt) make it difficult for SNV discrimination. In addition, the high cost and complex operational protocols severely limit their wider use and applications, especially in resource-limited settings. Most recently, a heteromultivalent DNA-functionalized particle assay[20] demonstrated single-base resolution in detecting viral RNA by utilizing spacer lengths and binding orientation. However, the model for fine-tuning binding affinity in this assay is complex, and it may struggle to maintain effective SNV detection when the SNV is situated in the central region of the miRNA sequence. Other current technologies for miRNAs detection are northern blotting[21], toehold-based strategies[22], microarrays[23], NanoString[24] and duplex-specific nucleases (DSNs)[25]. While they can achieve SNVs identification, they have limited sensitivity, complex analytical workflows, or inherent trade-off between specificity and sensitivity in detecting low-frequency SNVs[26,27]. Hence, there is a need for facile and robust tools that can detect miRNAs with single-nucleotide specificity.

Among the existing SNV-based approaches, the clustered regularly interspaced short palindromic repeats (CRISPR) and CRISPR-associated (CRISPR/Cas) techniques have the potential to fulfill these unmet needs by using programmable guide RNA and collateral cleavage activity of Cas enzymes. Due to these advantages, CRISPR/Cas based assays have been widely utilized in various fields for RNA detection[28–31]. For example, the reported EXTRA-CRISPR method[32], which is a one-pot isothermal assay that utilizes rolling-circle amplification (RCA) along with the Cas12a enzyme, achieves SNV detection of miR-21 with a limit of detection at 1.64 fM. However, it achieves reasonable specificity, with a non-specific signal level below 25%, only when the SNV is positioned within a narrow detection range of just 3 nucleotides from the ligation point. This limited range is inadequate for effectively covering the full length of miRNA sequences (~22 nt). Deliberately incorporated mismatches can be introduced in the crRNAs, which enable Cas13a to successfully distinguish RNA alleles differing by a single nucleotide[33,34]. The study revealed a generic guideline (a mutation in position 3 of the spacer and the introduced mismatch in position 5)[34] that would confer Cas13a the optimal specificity. However, adhering to this basic guideline is not always possible due to the targeting miRNAs being short (~22 nt) and the high sequence similarity between mutant alleles within same family. This makes the effective discrimination of SNVs in miRNAs using Cas13a a challenge. Therefore, the assay discriminating SNVs via purposely introducing synthetic mismatch is only applicable to miRNAs with limited mutated positions[34]. Additionally, such approaches often compromise sensitivity in order to enhance specificity. Here, we aimed to develop a universally applicable CRISPR/Cas-based method that would allow us to directly detect target miRNA alleles with any single-nucleotide mutation at any position and with high sensitivity.

To balance the specificity for SNV discrimination and the sensitivity for miRNA detection, we presented an approach called peptide nucleic acid (PNA) mediated CRISPR/Cas13a system (PRICE). PRICE was designed to achieve the detection of SNVs in miRNAs via the elimination of the "false positive" signal resulting from the binding of non-target miRNA to the Cas13a/crRNA complex. Figure 1 provides an overview of the PRICE technique and its functionality. To determine the target miRNA with SNV at a specific locus of interest (or a wildtype miRNA), a panel of PNAs that can hybridize non-target miRNAs with SNV at other loci will be added to a sample containing a mixture of target and non-target miRNAs (Fig.1b). All PNAs will hybridize with non-target miRNAs except the target miRNA and thus cannot bind to the crRNA in the Cas13a/crRNA complex (Fig.1a). However, the unhybridized target miRNA will hybridize with crRNA in the Cas13a/crRNA complex and thus activate the Cas13a to cleave the RNA linker between a fluorescent reporter and quencher, allowing the fluorescence detection for reporting the presence and amount of the target miRNA.

In a patient sample, which may contain both mutant miRNA alleles and wild type miRNA, PNAs are designed to specifically target the mutant alleles. The utilization of PNA offers improved sensing specificity, primarily due to two distinctive properties[35,36]. (1) The uncharged backbone of PNA could significantly enhance the binding affinity of PNA/RNA (or DNA) complexes compared to their DNA/RNA (or DNA) counterparts by eliminating the electrostatic repulsion caused by the two negatively charged nucleic acid backbones, particularly under conditions of salt concentration below 100 mM; and (2) even a single base mismatch in PNA/RNA (or DNA) heteroduplexes could result in a pronounced reduction in the melting temperature (8–20 °C). These PNAs with perfect complementarity to the mutant alleles can prevent the binding of mutants to crRNA through Watson-Crick hybridization.

In PRICE, the PNA molecule was employed to eliminate the interference from mutant alleles, thereby allowing the Cas13a/crRNA complex to directly target the miRNA of interest and achieve the identification of SNVs. Due to the superior binding affinity of PNA/RNA (or DNA) complexes in full match and a significant drop of melting temperature with even a single-base mismatch (melting temperature could drop 20 °C for a PNA/DNA complex with a single-base mismatch)[36], these inherent properties of PNA in PRICE enable precise customization of binding conditions, allowing strict binding of PNA to fully matched nucleic acid sequences while excluding those with one or more mutated nucleotides. This is crucial in ensuring the precise identification of SNVs in miRNAs. Additionally, the direct detection of miRNAs by PRICE, without the need for reverse transcription, synthetic mismatches in crRNAs, or massive screening of SNVs in miRNAs, thereby mitigating potential quantitative biases and simplifying operational procedures. To demonstrate the simplicity and universal single-nucleotide specificity, we used PRICE to detect the let-7 family of miRNAs, which have highly similar sequences and play important regulatory roles in cancer[37,38], to validate its efficacy. Our findings suggest that PRICE holds promising potential for SNVs sensing in miRNA in clinical diagnostics.

## Results

### Assessing the feasibility of PRICE

To evaluate the feasibility of the PRICE, we first conducted a systematic assessment of the specificity of LbuCas13a by examining its performance against two sets of mutants harboring single-base mutations (Supplementary Fig. 1). The first set of mutants, which encompassed various single-base substitutions (labeled in red) along the 22-nt SNV-sensitive region of the target miRNA, while the second set of mutants comprised different single-base substitutions (labeled in blue) at the same positions located at two sites of the mutants. The results showed poor specificity of LbuCas13a for discrimination sequences containing SNVs, consistent with previous studies[34,39]. To achieve the SNV sensing resolution, the commonly used methods involve introducing a mismatch to the crRNA, which may require extensive experimental selection for optimal specificity, especially for target miRNA. However, in PRICE, PNA can be easily designed to efficiently eliminate false positive signals from mutant alleles. This approach only allows the

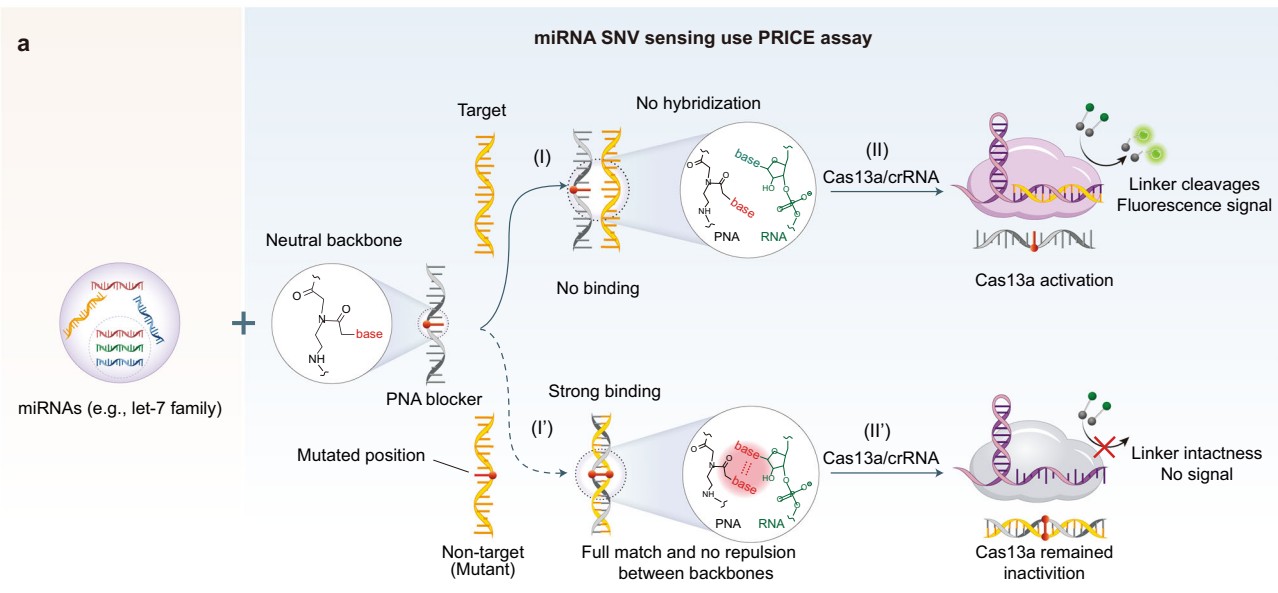

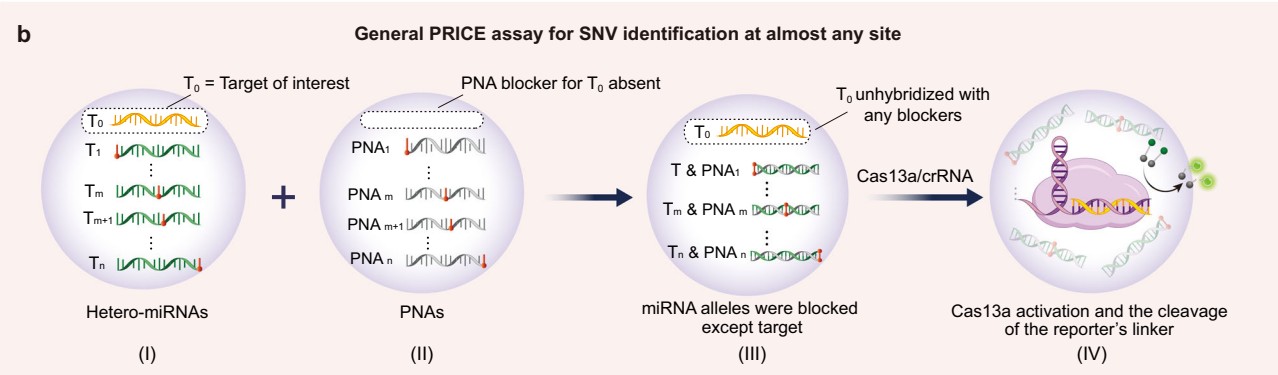

**Fig. 1 | Schematic of PNA mediated CRISPR/Cas13a system (PRICE) for miRNA detection with single base resolution. a** Single nucleotide variant (SNV) detection was accomplished by utilizing PNAs, specifically designed for hybridization with non-target miRNA (e.g., with mutant alleles compared to the target) (I' and II') but not with the target miRNA (I and II). In the presence of target miRNA, the PNA will not bind the target to form a duplex (I), allowing the target to bind with the crRNA in the Cas13a/crRNA complex to activate the Cas13a to cleave the link between the fluorescent reporter and quencher and thus initiating the fluorescence signal (II). However, the PNA will bind to the non-target miRNA to form a duplex (I'), preventing the binding of the non-target with the crRNA in the complex to activate the Cas13a and suppressing the fluorescent signal generation (II'). Consequently, individual members of the let-7 family of microRNAs (miRNAs) can be distinctly identified. **b** A general schematic of PRICE strategy for SNV identification across the whole miRNA sequence. (I) a schematic representation of the target ($T_0$) alongside its corresponding mutants ($T_1$ to $T_n$), wherein the single nucleotide mutations are positioned at various locations throughout the entire target sequence; (II) the designed PNAs labeled as $PNA_1$ to $PNA_n$, specifically targeting all mutant variants while excluding the target of interest, $T_0$ (the gray dashed frame showed without adding PNA for target $T_0$ miRNA); (III) mutants were blocked via hybridizing with PNAs except the target of interest $T_0$; (IV) only the target of interest $T_0$ could engage with and activate the Cas13a/crRNA complex, resulting in the cleavage of the linker between the fluorescent reporter and quencher and thus generating a fluorescent signal.

target miRNA to initiate the enzymatic activity of the Cas13a/crRNA complex, triggering the subsequent signaling and enabling SNV identification. To test the performance of PRICE, three mutants (10 A → G, 12 G → U, and 19 A → C) with mutated positions spanning the non-mutant target region (let-7a) were selected. The mutants were pre-incubated with their corresponding PNA probes, followed by mixing with the Cas13a/crRNA complex and signaling measurement. As depicted in Fig. 2a–f, the addition of PNAs, which formed Watson-Crick hybridization complexes with the mutants, significantly reduced the "false positive" signal (by more than 70%). Importantly, the presence of PNA molecules did not interfere with the signal originating from the target miRNAs (Supplementary Fig. 2), nor with other non-target nucleic acids (Supplementary Fig. 3).

To further demonstrate the enhanced capability of the PRICE assay to distinguish SNVs, a commonly used method that introduces a mismatch in the crRNA for the CRISPR/Cas13a system was used as a

comparative benchmark. The synthetic mismatch was strategically positioned at nucleotide 5 of the crRNA, based on previous studies[34], while the mutated positions for the eleven distinct mutants spanned almost the entire miRNA sequence. As illustrated in Fig. 2g, the PRICE assay exhibited superior performance in identifying miRNA SNVs compared to the synthetic mismatch crRNA method. Specifically, 5 ($T_7$, $T_9$, $T_{13}$, $T_{15}$, and $T_{21}$) out of 11 mutated positions showed specificity below the acceptable threshold ("false positive" signals below 30%) using the mutated crRNA sensing approach, whereas only 1 ($T_1$) out of 11 positions fell above this threshold when using the PRICE method. A similar trend was observed when comparing PRICE to the EXTRA-CRISPR technique: 3 out of 6 mutated positions showed specificity below the acceptable threshold with EXTRA-CRISPR, while 5 out of 6 positions maintained specificity below the threshold using PRICE (Supplementary Fig. 4). These results confirm that the PRICE assay significantly improves the biosensing specificity

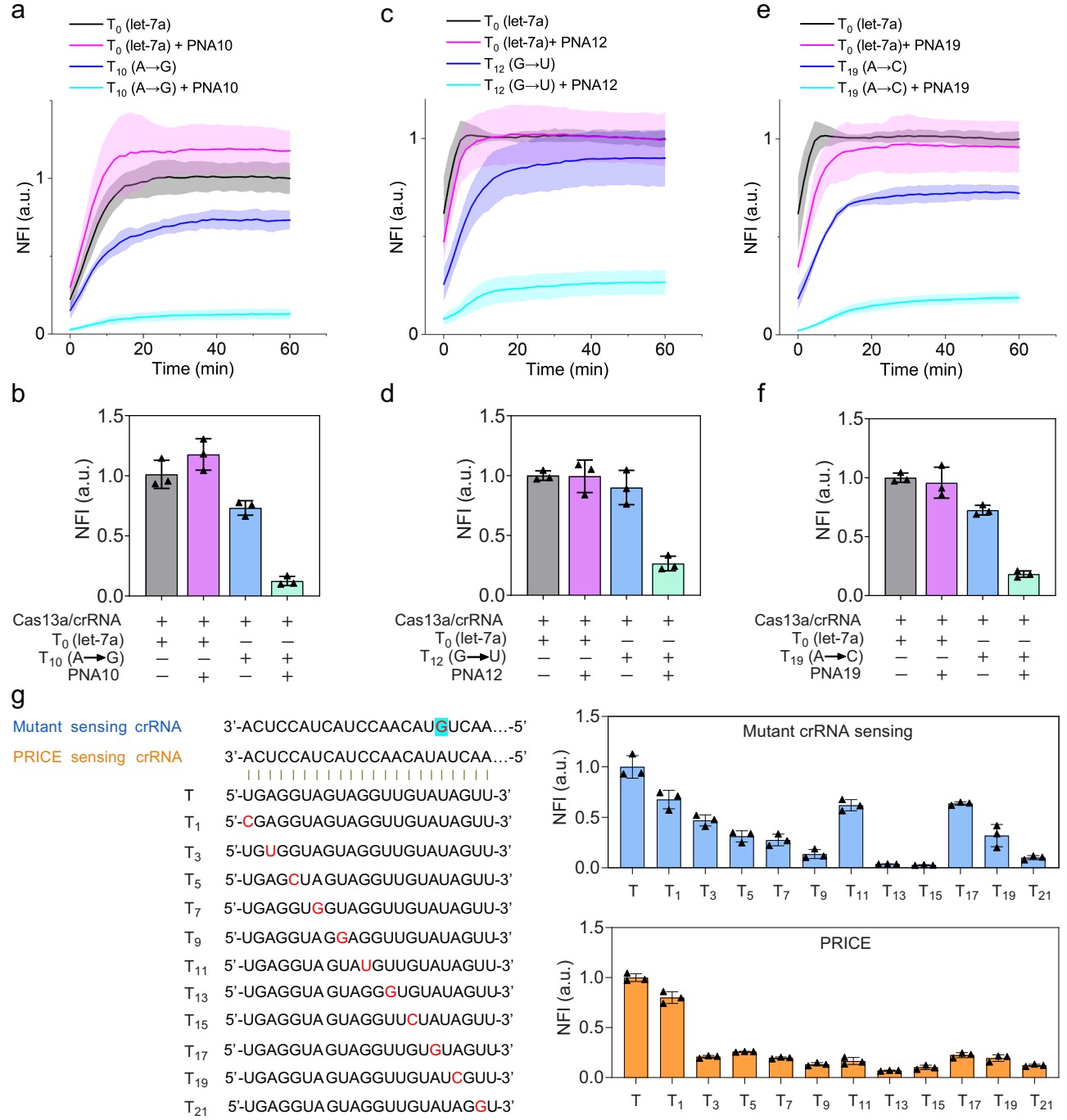

**Fig. 2 | Specificity of the PRICE assay for the sensing of miRNAs and its enhancement by using PNA. a–f** Characterization of PRICE assay for sensing mutants (10 pM) ($T_{10}(A \rightarrow G)$ (**a, b**), $T_{12}(G \rightarrow U)$(**c, d**), and $T_{19}(A \rightarrow C)$(**e, f**). **a, c,** and **e,** kinetics measurements, **b, d,** and **f,** 1h endpoint measurement of **a, c,** and **e,** respectively. **g** A comparative analysis of specificity between the PRICE assay and a conventional Cas13-based method[34] that involves the use of introduced mismatched bases in the crRNA to detect target miRNA and its mutants (10 pM) at varying sites across the sequence spectrum. Reactions were conducted with 10 nM PNA. Data represent the mean ± s.d., $n = 3$ technical replicates. The red letter in the figure refers to the position of the mutated base in the sequence.

of Cas13a and holds promise for the detection of SNVs in target miRNAs.

## Improving PRICE capacity to detect SNVs

An ideal PRICE assay should fulfill two key requirements: (i) effective prevention of the binding of mutants to crRNA to eliminate "false positive" signals, regardless of the presence or absence of the target, and (ii) ensure that the addition of PNA does not interfere with the detection of the target nucleic acid. To optimize the performance of PRICE, we investigated the influence of the PNA sequence length (or

$T_m$) on the sensing of both the non-mutated target miRNA and its mutant. The addition of PNAs designed to match the mutant miRNA fully, had the potential to interfere with the sensing of the target through hybridization between the target and the PNA molecule, given that there was only one SNV between the target and mutant. As expected, we observed a gradual increase in interference for the detection of the non-mutant miRNA (let-7a) as the length of the PNA increased by one base each time, ranging from 7 to 9 base pairs, with corresponding melting temperatures for PNA/target ($T_{m\_mis}$) of 22.0 °C, 28.3 °C, and 32.2 °C, respectively. As demonstrated in Fig. 3a,

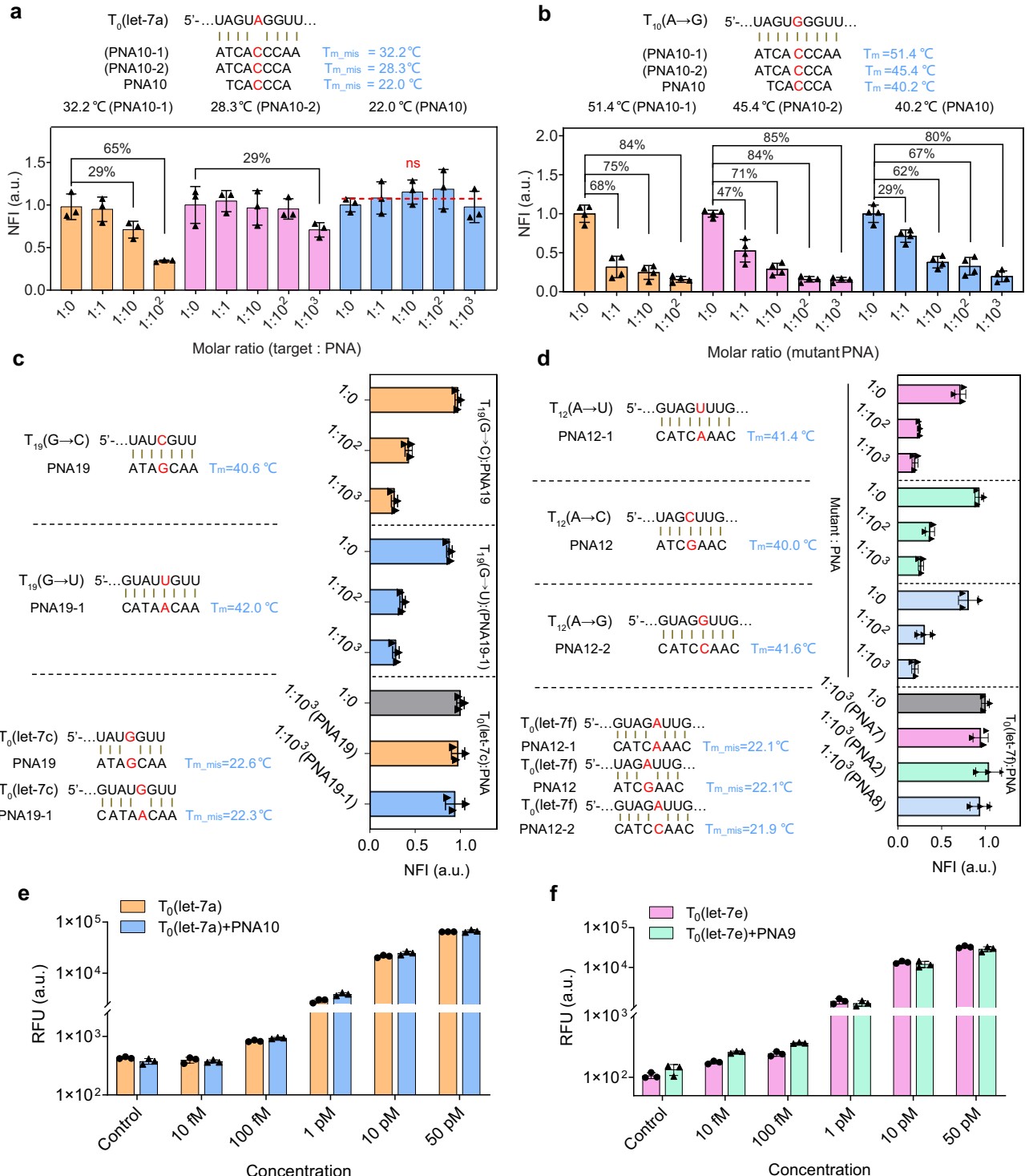

**Fig. 3 | Optimization of the PRICE assay for the sensing of miRNAs. a, b** The impact of melting temperatures of PNAs with target (**a**) or mutant (**b**) for the performance of PRICE. The molar ratio of target (10 pM) to PNA (**a**) or mutant (10 pM) to PNA (**b**) varied among 1:0, 1:1, 1:10, 1:100, and 1:1000. **c** Validation of optimized PRICE for sensing let-7 family mutant $T_{19}(G \to N)$ (10 pM). **d** The Specificity of optimized PRICE for sensing let-7 family mutant $T_{12}(A \to N)$ (10 pM). **e, f** Sensing target of let-7a (**e**) or let-7e (**f**) at different concentrations in the presence and absence of PNA10 (**e**) or PNA9 (**f**), showing PNA10 or PNA9 does not affect the detection limit of let-7a or let-7e. The let-7e was pre-diluted with 20 mM Tris-HCl, pH 7.5, subjected to extraction and enrichment using a magnetic nanoparticle method prior to measurement. The concentration of PNA used was 1000-fold higher than that of the sensing targets. The control represents the background signal in the absence of the target miRNA. Data represent the mean ± s.d., $n = 4$ technical replicates for (**b**) and $n = 3$ technical replicates for the rest. The red letter in the figure refers to the position of the mutated base in the sequence.

the signal interference of PNA5 (8 bp with $T_{m\_mis}$ = 28.3 °C) for the non-mutated target was 29% at a molar ratio of non-mutant to PNA of $1{:}10^3$, which increased to 65% for PNA10-1 (9 bp with $T_{m\_mis}$ = 32.2 °C) at a molar ratio of non-mutant to PNA of $1{:}10^2$. However, there was almost no interference for PNA10 (7 bp with $T_{m\_mis}$ = 22.0 °C), even at a higher molar ratio of target to PNA of $1{:}10^4$ (Supplementary Fig. 2). Additionally, for the inhibition of the "false positive" signal from the mutant, as shown in Fig. 3b, although the signal decreased larger and larger as PNA length varied from 7 bp to 9 bp at a molar ratio of mutant to PNA of 1:1 (the signal inhibition rate increased from 29% at 7 bp to 68% at 9 bp), a different scenario emerged at high molar ratio of PNA to mutant, where the "false positive" signal inhibition decreased gradually and maintained at approximately 80% at a molar ratio of mutant to PNA of $1{:}10^3$. In other words, very short PNAs failed to fully block the mutant, while very long PNAs increasingly interfered with wild-type miRNA detection (Supplementary Fig. 5 and Supplementary Table 1). Therefore, to maximally inhibit the "false positive" signal generated from mutant and maximally avoid interference from PNA molecules in the target, we conclude that the design of PNA molecule should ideally adhere to the following two guidelines: (1) the melting temperature of PNA/target ($T_{m\_mis}$) should not exceed 22.0 °C, with a lower $T_{m\_mis}$ being preferable for PRICE, and (2) the melting temperature of PNA/mutant ($T_m$) should not be lower than 40.0 °C.

To validate the two guidelines, we measured *let-7c* and its two mutant alleles, $T_{19}(G \to C)$ and $T_{19}(G \to U)$. As shown in Fig. 3c, the $T_{m\_mis}$ of PNA19/let-7c was 22.6 °C, and the $T_m$ of PNA19/ $T_{19}(G \to C)$ was 40.6 °C, while the $T_{m\_mis}$ of PNA19-1/let-7c was 22.3 °C, and $T_m$ of PNA19-1/ $T_{19}(G \to U)$ was 42 °C. All of these PNA molecules exhibited a "false signal" inhibition rate of over 70% for their corresponding mutants, while showing no interference in the detection of the target at the molar rate of mutant (or target) to PNA of $1{:}10^3$. The guideline is also applicable for the design of probes targeting let-7f mutants, as demonstrated in Fig. 3d, wherein the SNVs are situated at the 12th position of the sequences. These results further demonstrate the generality of the pipeline and confirm the importance of the proposed guidelines for the design of PNA probes.

We further optimized key parameters that could potentially impact the performance of PRICE, including buffer type, pH, and salt concentration (Supplementary Figs. 6–7). The results showed that the choice of buffer (Tris-HCl or HEPES) had little effect on performance. However, an optimal pH of around 8.5 and a low salt concentration between 0 and 50 mM were preferred for the best results.

We assessed the limit of detection (LOD) using both Cas13a and PRICE for the sensing of let-7a and let-7e with a PNA concentration 1000-fold higher than the target. For let-7e, it was pre-diluted with 20 mM Tris·HCl pH 7.5 solution, subjected to extraction and enrichment using a magnetic nanoparticle method prior to measurement (Supplementary Figs. 8, 9). The PRICE assay demonstrated a LOD of 24 fM for sensing let-7a, which is almost equivalent to the Cas13a/crRNA complex alone at 19 fM (Fig. 3e, Supplementary Fig. 10). A similar level of LOD ($\sim$10 fM) for sensing let-7e was also achieved by the PRICE assay (Fig. 3f). This finding is consistent with previous studies on miRNA detection, which have reported LOD values ranging from 1 to 10 fM[32,40]. This level of sensitivity enables direct detection of specific miRNA, such as let-7 family miRNA, in clinical tissue samples[41,42] (the standard amount of let-7e and let-7f are both within the range of $3 \times 10^6$ to $1.5 \times 10^7$ copies/mg in liver tissue). This finding provides further evidence that the addition of PNA molecules does not interfere with the PRICE's ability to detect miRNA targets.

To validate the performance of PRICE in the lower concentration range (such as fM), we measured both the target (let-7a) and its mutant (10 A → G) at 100 fM. As illustrated in Supplementary Fig. 11, the addition of the PNA blocker in the PRICE assay effectively suppresses the false signal from the mutant and enhances assay specificity.

Lastly, we established a predictive model ($Tm_{(PNA/RNA\_mis)}$), using the nearest neighboring dimers method, to guide the design of PNA blockers. For effective detection of SNVs, the $T_{m\_mis}$ between PNA and the target miRNA should be significantly lower than the assay's measuring temperature of 37 °C. Conversely, the $T_m$ between PNA and the mutant allele should be higher than 37 °C, as outlined in the guidelines. These data (Fig. 2a–g, Fig. 3a–d, Supplementary Figs. 12, 13, Supplementary Data 1, Supplementary Table 2 and 3) also validate the accuracy and usability of the model we built. Thus, the predictive model can be used to improve the PRICE assay by guiding the design of PNA to achieve SNV detection.

## PRICE can be used for sensing miRNA in a mixture

To assess the ability of PRICE to detect SNVs in clinical samples, the specificity of this assay was evaluated using a mixed sample containing both the non-mutant miRNA (let-7a) and its mutant allele $T_{10}(A \to G)$. To mimic real clinical settings, separate measurements were taken for the pure non-mutant and mutant allele, as controls for comparison. The schematic of the PRICE assay for sensing this mixture is shown in Fig. 4a. While an overlapped signal could be produced by the target and its mutant allele, leading to a false signal and posing challenges for quantification, the PRICE assay effectively addressed this issue. The overlapped false positive signal was observed, but it gradually decreased until it ultimately converged with the authentic target signal as the PNA concentration varied from 2 pM to 2 nM (Fig. 4b, c). The results indicate that PRICE could be used for sensing miRNAs with SNVs in a heterogeneous environment.

Additionally, we sought to determine if the PRICE assay could detect low-frequency non-mutant miRNA targets in samples with mock mutant alleles, which is crucial because low levels of the miRNA target of interest (mutants) could exist within high levels of wild-type sequences. We found that PRICE was able to detect SNV-containing miRNA targets at levels as low as 5% of the non-mutant/mutant mixture samples (Fig. 4d).

Finally, in real sample detection, multiple mutant alleles could be present, for instance, the *let-7* family[37]. Therefore, it may be desired to design a single PNA sequence to eliminate interference from multiple mutants. To achieve this, we designed PNA9 (Fig. 4e) aimed to eradicate the interference from let-7a, let-7c, or let-7f, while sensing the target let-7e. The results (Fig. 4f, g, Supplementary Fig. 14) demonstrated an approximate 80% "false positive" signal inhibition rate for the mutants, while the PNA molecule had almost no impact on the sensing of the target. This suggests that a single PNA sequence can effectively "lock" multiple mutants, thereby improving the specificity of the assay.

## PRICE on the analysis of miRNA regulation from clinical samples

To further validate the effectiveness and clinical applicability of the PRICE assay, we conducted an evaluation of let-7e and let-7f miRNAs regulation in cell and human liver cancer tissue samples. Considering that the expression of miRNAs can vary across different cell types and serve as an indicator for miRNA regulation in cancer patients, we initially measured let-7e and let-7f miRNAs in the HepG2 cell line ($n = 5$), using the THLE-2 cell line ($n = 5$) as a control[43,44]. Total RNA extraction and purification were performed using a commercial kit, and an equal number of cells were utilized for each sample to ensure accurate comparisons. The same protocol was followed for the subsequent liver cancer tissue samples. The samples were analyzed in parallel by RT-qPCR and PRICE. In RT-qPCR, primers designed based on previous studies[45] were used. As depicted in Fig. 5b, c and Supplementary Fig. 15, the PRICE assay explicitly revealed down-regulation of let-7e and let-7f miRNAs in all HepG2 ($n = 5$) and THLE-2 ($n = 5$) cell samples, which was in agreement with the qPCR measurements and previous studies[46–49].

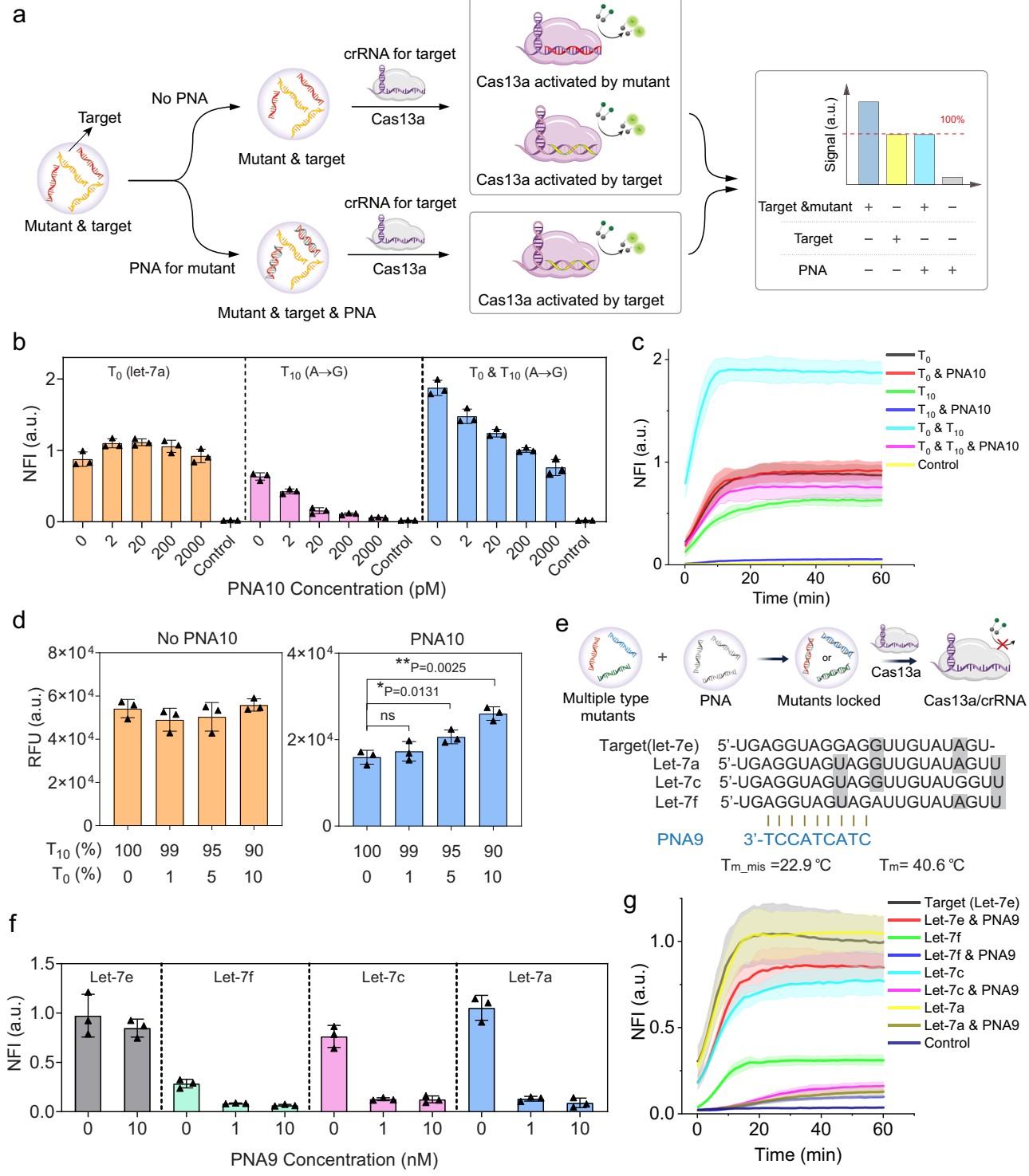

**Fig. 4 | The performance of the PRICE assay for the detection of miRNAs from a mixture. a** Schematic of PRICE for sensing miRNA in the mixture of target and mutant. The 100% red dashed line represents the signal from the target. **b** PRICE detection of a target miRNA from a solution containing this target (2 pM), its mutant $T_{10}(A \rightarrow G)$ (2 pM), or a mixture of both target and its mutant with each of 2 pM. **c** Kinetic measurements of the PRICE assay for sensing miRNAs corresponding to the data of (**b**). **d** PRICE can detect the trace amount of the target miRNA from a mixture of the target and its mutant $T_{10}(A \rightarrow G)$ at different molar percentage (paired two-tailed Student's t test, *$P < 0.05$, **$P < 0.01$). **e, f** Schematic (**e**) and

detection results (**f**) showing one type of PNAs (PNA9) can block multiple mutants (let-7a, let-7c and let-7f) when let-7e was viewed as a target, allowing us to detect let-7e from the solution (10 pM) of let-7a, let-7c, or let-7f using the PRICE assay. **g** Kinetic measurements of the PRICE assay were analyzed based on the data obtained from the corresponding (**f**). The control group in this study encompassed measurements taken in the absence of the target or mutant sequences. The column data were plotted using the 1-hour endpoint values derived from their respective kinetic measurements. Data represent the mean ± s.d., $n = 3$ technical replicates.

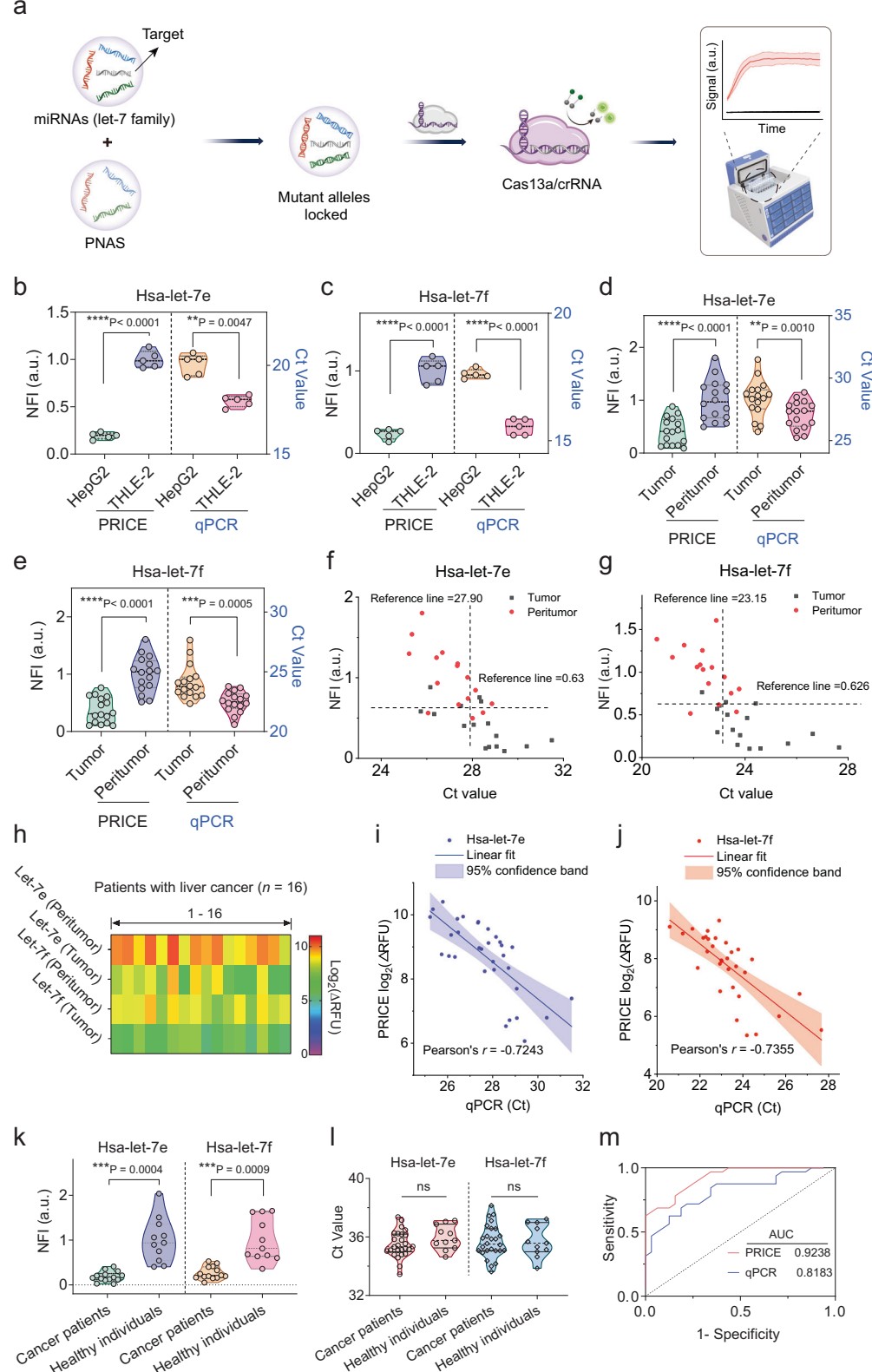

Subsequently, we extended our analysis to include human liver cancer tissue samples, with peritumoral tissue samples serving as controls obtained 1 centimeter away from the tumor lump in each patient. These samples from liver cancer patients ($n = 16$), collected between 22 February and 28 June 2023, were provided by Haikou People's Hospital. As mentioned above, the same amount of tissue was used for total RNA extraction in each sample, and the final PNA

concentration was set at 1 nM for PRICE. Both the PRICE and qPCR generated consistent results across all 16 samples, revealing the downregulation of let-7e and let-7f in patients with liver cancer (Fig. 5d−h, Supplementary Fig. 16 and Supplementary Table 4), in line with the previous studies[46–49]. To validate the PRICE measurements, the same data were analyzed using Pearson's $r$ value. An acceptable correlation was observed between the PRICE and qPCR methods

**Fig. 5 | PRICE analysis for the miRNA regulation in cell cultures, liver cancer tissue, or serum. a** Schematic representation of the sample analysis experimental workflow. **b, c** The expression levels of Hsa-let-7e (**b**) and Hsa-let-7f (**c**) in THLE-2 and HepG2 cell lines were measured by PRICE and qPCR methods, respectively (paired two-tailed Student's t test, **$P < 0.01$, ****$P < 0.0001$). **d, e** The expression levels of Hsa-let-7e (**d**) and Hsa-let-7f (**e**) in liver tissue biopsies from patients diagnosed with liver cancer were measured using the PRICE and qPCR techniques, respectively. The control samples were the peritumoral tissue located 1 centimeter away from the tumor mass, while the liver cancer samples were derived from the tumor mass tissue within the same patient (paired two-tailed Student's t test, **$P < 0.01$, ***$P < 0.001$, ****$P < 0.0001$). **f, g** Correlation scatter plots of *P*RICE and qPCR demonstrate the regulation of Hsa-let-7e (**f**) and Has-let-7f (**g**) from tissue samples, based on the data presented in d and e, respectively. **h** Heatmap of the expression levels of individual miRNAs (hsa-let-7e and hsa-let-7f) in extracted tissues from patients ($n = 16$) with liver cancer. **i, j** Correlation between the parallel measurements by PRICE and qPCR for hsa-let-7e (**i**) and hsa-let-7f (**j**). The data points represent the mean of at least three replicates of each measurement by each method. PNA concentration was fixed at 1 nM. **k, l** The expression levels of Hsa-let-7e and Hsa-let-7f were measured using the PRICE assay (**k**) and qPCR technique (**l**) in serum samples obtained from patients diagnosed with liver cancer as well as in samples from healthy individuals (paired two-tailed Student's t test, ***$P < 0.001$). The PNA concentration was fixed at 100 pM. **m** Comparative analysis of miRNA detection using the PRICE and qPCR assays through ROC curve analysis and calculation of the AUC for regulatory assessment in liver cancer. Data represent mean ± s.d., $n = 3$ technical replicates.

(Fig. 5i, j), with Pearson's *r* of -0.7243 for let-7e and Pearson's *r* of -0.7355 for let-7f, respectively.

Additionally, we applied the PRICE assay to analyze serum samples, which included 26 positive samples from liver cancer patients and 11 samples from healthy individuals serving as negative controls. Due to the low concentrations of target miRNAs in serum, let-7e and let-7f miRNAs were extracted and enriched using a magnetic nanoparticle method prior to measurement. The results indicated that while minimal signal was detected from the positive samples, the PRICE assay clearly demonstrated the downregulation of let-7e and let-7f in the serum of liver cancer patients (Fig. 5k, l, Supplementary Fig. 17 and Supplementary Table 5), while the qPCR analysis showed indistinguishable for regulation between the patents and healthy individuals. This may attribute to both the low levels of miRNAs (Ct values ~ 35).

Furthermore, the PRICE assay exhibited superior miRNA measurement performance in clinical patient samples compared to qPCR (Fig. 5d, e). The ROC curves (Fig. 5m, Supplementary Fig. 18) also validated this conclusion, as the PRICE showed an AUC of 0.9238 (0.9453 for let-7f, 0.9023 for let-7e) with a sensitivity and specificity of 81%, while the qPCR method exhibited an AUC of 0.8183 (0.8711 for let-7f, 0.7656 for let-7e) with a sensitivity and specificity of 75%.

While these clinical validations demonstrated high discriminatory power for the PRICE system, future large-scale, prospectively collected studies with more advanced normalization methods (e.g., standardization to total RNA content)[50,51] will be essential to confirm and refine these findings.

In summary, both the developed PRICE assay and qPCR revealed the downregulation of let-7e and let-7f in liver cancer patients, with the PRICE assay demonstrating significant potential for miRNAs detection in clinical settings. Meanwhile, to enhance the accessibility of the PRICE assay for numerous applications that necessitate portability and expedited results, we subsequently designed and developed a 16-channel portable fluorescence reader with a touchscreen interface specifically for the PRICE assay (Supplementary Fig. 19). The functionality of the PRICE instrument was validated by the measurement of let-7a miRNA using the PRICE assay with target concentrations varying from 50 pM to 50 fM (Supplementary Fig. 19). The results showed consistency with the previous data (Fig. 3e, f), indicating the superior performance of the PRICE device. As a proof of concept for clinical applications, we then conducted an analysis for the miRNA regulation in liver cancer tissue samples ($n = 15$) as well as serum samples using the PRICE device. As shown in Supplementary Fig. 20a–c (Supplementary Table 6), the PRICE device was able to detect the differential levels of let-7e and let-7f compared to the control group, which is consistent with the qPCR results and previous data (Fig. 5d, e). The same conclusion could be drawn from the ROC curves (Fig. 5m, Supplementary Fig. 20d). The collective findings affirm the viability and precision of the PRICE device, suggesting its potential integration with the PRICE assay.

## Discussion

In this work, we presented PRICE, an assay that enables direct and universal discrimination of miRNAs with SNVs at nearly any target position. The key factor enabling the detection of SNVs in this assay is the design of PNA, which can be varied in terms of length and sequence. Significantly, PRICE demonstrates effective identification of SNVs in miRNA without requiring any screening tests when switching to new sensing targets. It is noteworthy that complete elimination of the "false positive" signal from mutants can be accomplished when the mutant possesses multiple base variants. Besides, the PRICE assay effectively inhibits false positive signals from mutants and successfully detects the real positive signal of the target in a mixture of target and mutant alleles. Also, the PRICE assay can directly detect miRNAs without a pre-amplification process. This assay can potentially be used for the detection of a variety of pathogens and diseases with SNV at almost any site, though the SNV detection by PRICE is diminished when mutated positions are located at the two distal of the miRNA sequences (approximately 30% inhibition rate for mutants) (Supplementary Fig. 21). To overcome this limitation, several complementary strategies can be employed: (1) optimizing guide RNA design using machine learning and generative algorithms like BADGERS[52]; (2) engineering the Cas enzyme through amino acid substitutions or directed evolution methods[53–55], including phage-assisted continuous evolution; and (3) chemically modifying the PNA backbone, for example, by introducing lysine, alanine, or glutamic acid at the γ-carbon position. Integrating these approaches offers a promising path to enhance PRICE's performance and address its current constraints.

In addition to SNVs in RNAs, PRICE exhibits universal discrimination of SNVs in DNA, allowing for highly specific detection of DNA mutant alleles. We utilized PNA to block DNA mutant allele in CRISPR/Cas14a to inhibit the "false positive" signal (Supplementary Fig. 22), since Cas14a can also tolerate single-base mismatched nucleic acid sequences, particularly when the mutation is located at two sides of its sensing target sequence (Supplementary Fig. 23). Our study demonstrated that increasing the molar ratio of PNA to mutant or target led to a higher inhibition rate of the false positive signal, or a lower interference rate for the signal of the target. When the molar ratio of PNA to mutant (or target) exceeded 100-fold, the false positive signal generated by the mutant was completely inhibited, while the interference rate of the PNA for the sensing of the target was close to zero (Supplementary Fig. 22). This assay significantly improved the specificity of Cas14a for nucleic acid sensing. The specificity of Cas14a in more complex samples was further studied by detecting low-frequency targets in the presence of a high amount of mutant allele. We showed that the use of PNA enables clear differentiation of various target concentrations, while its absence does not (Supplementary Fig. 24). These results highlight that the substantial improvement in specificity is achieved by the incorporation of PNA into Cas14a.

Although PRICE enabled the identification of SNVs in miRNA from tissue samples, its sensitivity remains inadequate for detecting

samples with low levels of target miRNA, such as circulating miRNA in blood. To address this limitation and enhance the LOD of PRICE, several approaches could be developed: (1) the acquisition of more sensitive Cas enzymes through structural biology or continuous evolution of phage[56,57] (e.g., reprogramming enzyme), (2) the integration of amplification elements or methodologies into the PRICE framework[31,58] (e.g., leveraging GFET or cascaded enzymes) and (3) target enrichment[29,59](e.g., magnetic nanoparticles or digital PCR).

The CRISPR-Cas13/Cas14-based PRICE assay has the potential to be applied beyond its current scope, using programmable PNA to integrate with other platforms that improve SNV capacity, such as homomultivalent[20], molecular logic systems[60], and toehold-mediated switch[61]. In genetic diagnostic tools, the powerful binding ability that allows excellent, specific, and precise binding is essential, making PRICE a potential tool for further developing clinical diagnostics. Additionally, PNA probes confer enhanced specificity and stability through strong mismatch discrimination (8 ~ 20 °C $T_m$ drop per mismatch), salt tolerance, and nuclease resistance[35,36,62], while the sequence-specific activity of well-characterized Cas proteins further improves selectivity in complex biological matrices[32,63,64]. Together, these attributes render the PRICE platform robust to interferents such as non-target RNA and DNA and make it a promising candidate for clinical diagnostics; moreover, a comparative analysis of detection time, sensitivity, and cost (Supplementary Tables 7–8) indicates that PRICE is practical and competitive for clinical use.

The modular architecture of the PRICE system suggests significant potential for adaptation beyond miRNAs. Future work will explore its application to other clinically relevant RNA classes, such as snoRNAs[65] and lncRNAs[66]. Key considerations for this adaptation include the design of target-specific capture probes optimized for potentially lower abundance or structurally complex targets, and potential adjustments to sample processing protocols. Successful extension to these RNA types could significantly broaden the diagnostic and research utility of the platform.

## Methods

### Ethics statement

All the tissue samples were collected from Haikou People's Hospital in Haikou, China. The Ethics Committee of Haikou People's Hospital in Haikou, China (Permission SC20230052, 22 February 2023) approved all of the procedures, and all methods were performed in accordance with the relevant guidelines and regulations. All subjects included in this study signed an informed consent for tissue obtained for research purposes. Under a waiver of informed consent, we obtained the discarded excess serum samples from the Department of Laboratory Diagnostics, Changhai Hospital, Navy Military Medical University, Shanghai, China. Sex and/or gender were not considered in the study design because de-identified samples were used, and no demographic data (including sex/gender, age) were collected.

### Reagents and materials

All the nucleic acid primers and probes were synthesized by Sangon Biotechnology Co., Ltd (China). LbuCas13a was purchased from Bio-Lifesci (China), and Cas14a1(Plasmid #112500) expression plasmid was purchased from Addgene (USA). The pET-28a-TEV expression plasmid was purchased from NovoPro Bioscience Inc. (China). The trans BL21 (DE3) pLysS competent cell was purchased from TransGen Biotech Co., Ltd (China). PNA sequences were purchased from Panagene (Korea).

### Protein expression and purification

Cas14a1 protein expression and purification were performed as our previous work[67]. Briefly, the E.coli Rosetta (DE3) pLysS were transformed with pLBH531_MBP-Cas14a1 and pET-28a-TEV expression plasmids, respectively, and grown in Luria broth (LB) media containing 50 µg/mL ampicillin at 37 °C and 180 rpm until OD600 reaches 0.6. Afterwards, protein expression was induced with isopropyl-1-thio-β-D-galactopyranoside (IPTG) (0.1mM for TEV, and 0.2mM for Cas14a1) and cultured overnight at a certain temperature (30°C for TEV and 18°C for Cas14a1) and 180rpm. Cells were harvested by centrifugation at 10,000 x $g$ for 10 min at 4 °C, and lysed by sonication in the lysis buffer (20 mM Tris-HCl, pH 8.0, 500 mM NaCl, 1 mM DTT, 0.2 mM PMSF, 5% glycerin, 5 mM β-mercaptoethanol, 10 mM imidazole). Lysates were separated by centrifugation at 6700 x $g$ for 30 min under 4 °C, and the supernatant solutions were added into Bio-Scale Mini Nuvia IMAC Ni-Charged (Bio-Rad Laboratories, Inc., USA) and eluted with imidazole with linear gradient by increasing concentration (from 10 to 300 mM) in elution buffer (20 mM Tris-HCl, pH 8.0, 500 mM NaCl, 5 mM β-mercaptoethanol). The 10his-MBP-tag of Cas14a1 was removed by TEV protein with overnight incubation at 4 °C. The solution was further replaced with a buffer (20 mM Tris-HCl, pH 8.0, 100 mM NaCl, 5 mM β-mercaptoethanol) and purified by a heparin column (GE, Hi-Trap) with linear gradient elution of NaCl in concentration from 100 mM to 2 M. The purified products were analyzed by sodium dodecyl sulfate-polyacrylamide gel electrophoresis (SDS-PAGE) (Supplementary Fig. 25), and quantified with BCA Protein Assay Kit (Cat# PC0020, Beijing Solarbio Science & Technology Co., Ltd.). The proteins were collected in storage buffer (20 mM Tris-HCl, pH 8.0, 1 M NaCl, 50% glycerol) and stored at -80 °C until use.

### crRNA (or sgRNA) preparation

The crRNAs of Cas13a (sgRNA of Cas14a) were produced by in vitro transcription using T7 RNA polymerase (HiScribe® T7 High Yield RNA Synthesis Kit, NEB). Briefly, double stranded DNA templates for crRNAs that containing T7 promoter sequence were prepared by gradient cooling (holding at 95 °C for 5 min and then gradient cooled (5 °C min⁻¹) to room temperature). Then the transcription reaction was performed with DNA templates, T7 RNA polymerase NTP mix at 37 °C for 6 h. The final transcription RNA was obtained by using Monarch RNA Cleanup Kit (NEB), analyzed by polyacrylamide gel electrophoresis (PAGE) (Supplementary Fig. 26), quantified using Nanodrop 2000 (ThermoFisher) and stored at −80 °C until use.

Template for sgRNA was obtained by PCR. The reaction contained 2 µL primers F (1 mM), 2 µL primers R (1 mM), 25 µL premix TaqTM (Takara), 2 µL vectors (20 ng/µL) and adding nucleic acid-free water to 50 µL. Amplification program was followed: initial incubation at 95 °C for 5 min, then 30 cycles of denaturation at 95 °C for 30 sec, annealing at 54 °C for 30 sec and elongation at 72 °C for 30 sec, final extension at 72 °C for 10 min. The amplified product served as the DNA template for in vitro transcription at 37 °C for 6 h. The transcription product was purified by Monarch RNA Cleanup Kit (NEB), analyzed by polyacrylamide gel electrophoresis (PAGE) (Supplementary Fig. 26), quantified using Nanodrop 2000 (ThermoFisher) and stored at −80 °C until use.

The sequences of all DNA were listed in Supplementary Data 2.

### PRICE for the sensing of SNVs

The influences of buffer, pH, salt concentration and types of blockers for Cas13a reaction condition were optimized (Supplementary Figs. 6, 7 and Supplementary Fig. 27). The PNA molecules were prepared in 20% acetonitrile solution, then the solution was heated to 65 °C for 10 minutes and chilled on ice for five minutes. Next, it was mixed with nucleic acids at certain ratio and hybridized in solution containing 40 mM KCl for 1 h at 4 °C.

Cas13a/crRNA complexes were pre-incubated in 1×reaction buffer (10 mM Tris-HCl, 10 mM MgCl₂, 40 mM KCl, pH=8.2) at 37 °C for 15 min. The fluorescence measurement was performed with 20 nM purified LbuCas13a, 20 nM crRNA, 300 nM dual-labeled (FAM and BHQ1) RNA reporter probe and hybridized product in 1× reaction buffer on 384-well microplate. The fluorescent signals were monitored

in BioTek H1 microplate reader ($\lambda_{ex}$: 492 nm; $\lambda_{em}$: 520 nm) up for 1 h at 37 °C, and the dynamics were measured every minute.

Normalized fluorescent intensity (NFI) was the ratio of the net signal (obtained by subtracting the background from the measured signal) between the set of measurement objects and the normalized one; Normalized intensity (NI) was the ratio of the measured signal between the set of measurement objects and the normalized one; Relative fluorescent unit (RFU) was the measured signal; ΔRFU was the difference between RFU and the background fluorescence.

The RNA and PNA sequences were listed in Supplementary Data 3 and Supplementary Table 9, respectively.

## Cell culture
The THLE-2 cells (ATCC, CRL-2706 ™) were cultured in THLE-2 complete medium (purchased from ProCell), while HepG2 cells (Hzbscience, Cat# HZ-C0405) were cultured in DMEM complete medium (purchased from ProCell). All the cells were cultured at 37 °C in a 95% humidified atmosphere with 5% $CO_2$. When the cells reached 80–90% confluence, they were passaged using 0.25% trypsin-EDTA solution (purchased from Gibco). Cell counting was performed using the Venus Milo fully automated high-throughput cell counter.

## Serum miRNA extraction and enrichment using magnetic nanoparticles
The miRNAs were isolated using Streptavidin MagPoly Beads (Smart-Lifesciences, Cat# SM017001). Prior to extraction, the Streptavidin MagPoly Beads were functionalized with biotin-capture primers (the primer sequences are provided in Supplementary Data 2) in a 0.5 SSC buffer solution for 1 hour at room temperature. The beads were then washed and equilibrated in a 20 mM Tris-HCl buffer at pH 7.5 before utilization.

Next, the modified magnetic nanoparticles (MagPoly Beads) were mixed with 1 mL of serum and incubated at 4 °C for 30 minutes. Subsequently, the beads were washed with a 20 mM Tris-HCl buffer at pH 7.5 containing 0.5 M NaCl. The extracted miRNAs were eluted from the magnetic beads using DEPC water, and the final miRNA product was stored at −80 °C for future use (Supplementary Fig. 17).

## Detection of miRNAs by Real-time PCR
The let-7 family miRNAs from cells ($1 \times 10^6$ cell per sample) or patient tissue samples (70 mg per tissue sample) were extracted by using PureLink™ RNA Mini Kit (Invitrogen™, Cat# 12183018 A) and resuspended in deionized water. Then reverse transcription was performed according to the previous study using PrimeScript™ RT reagent Kit (Perfect Real Time) (Takara, Cat# RR037A) with 10 μM RT primers 1 μL, 1 μL extracted sample template, 2 μL 5X PrimeScript Buffer (for Real Time), 0.5 μL PrimeScript RT Enzyme Mix I, RNase Free $dH_2O$, totally 10 μL reaction solution. Program was run at 37 °C for 15 min, and inactivation of reverse transcriptase at 85 °C for 5 sec.

Subsequently, the real-time PCR was performed using 2xQ3 SYBR qPCR Master mix (Universal) (TOLOBIO, REF 22204-01) kit with 2 x Q3 SYBR qPCR Master Mix, 300 nM forward and reverse primer, 0.8 μL reverse transcription product, $ddH_2O$ up to 20 μL, and carried out on LightCycler 96 SW qPCR instrument. All the sequences of primers were listed in Supplementary Data 2.

## Constructing the PRICE device
The LED illuminator (C503B-BAN-CZ0A0451) and fiber optics (WSW-SC-01) were purchased from Wolfspeed, Inc. and Beijing OPTICS Optoelectronic Technology Co., Ltd., respectively. The PCBA circuit boards and PRICE device shell (WSW-SC-01) were purchased from Shenzhen JLC Technology Group Co., Ltd. The light signal from the sample is converted into photocurrent via Si photodiodes (S2386-44K, Hamamatsu Photonics), then transferred into the output digital signal via an analog-to-digital converter (AD7490, Analog Devices), and

displayed on a touchscreen module (RK3568, Beijing Marvsmart Technology Co., Ltd.). A PI heating film (Yancheng Zhenglong Electric Heating Technology Co., Ltd) maintains a 37 °C reaction temperature, and a motor (35BYG252-028A, Changzhou Yuanrui Motor Technology Co., Ltd.) moves the detection module for channel switching. These functional modules are controlled with a microcontroller (STM32F103RCT6, STMicroelectronics). The PRICE device is compatible with a 200 μL reaction tube for the reaction, along with 16 parallel measurements. The dimensions of the PRICE device: (length x width× height) $20 \times 15 \times 16$ cm.

## Statistics and reproducibility
All experiments and assays were repeated at least three times. The data were expressed as mean ± s.d. and compared by Student's t test. The GraphPad Prism version 8.0, and OriginPro 2021 were used for data analysis. No statistical method was used to predetermine sample size. No data were excluded from the analyzes. Allocation was not randomized (observational design). Experimental procedures were performed with blinding to the sample group.

## Reporting summary
Further information on research design is available in the Nature Portfolio Reporting Summary linked to this article.

## Data availability
The main data supporting the findings of this study are available within the paper and its Supplementary Information. All the original data generated in this study are provided as Source Data. Source data are provided with this paper.

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

## Acknowledgements

This work was financially supported by National Natural Science Foundation of China (No. 42376184 to Y.Wan, No.42306225 to Z.Y., No.42166001 to Y.Wan, No.41866002 to Y.Wan, No.82402756 to Y.Wei, No.42576187 to Y.Wan, No. 42576122 to F.S. and No. 42206212 to Y.Wan), Research Foundation of Hainan University (No. KYQD(ZR)-21128 to Y.Wan), International Science & Technology Cooperation Program of Hainan Province (GHYF2025015 to Y.Wan), Joint Program on Health Science & Technology Innovation of Hainan Province (WSJK2025MS135 to Y.Wan), Hainan Provincial Natural Science Foundation of China (No. 322MS032 to B.W.), the Central Government Guides Local Science and Technology Development Projects (No. ZY2022HN01 to Y.Wan), Hainan Science and Technology Talent Innovation Project (KJRC2025B17 to Y.Wan), Collaborative Innovation Center of Marine Science and Technology, Hainan University (No. XTCX2022HYC12 and No. XTCX2022HYB09 to Y.Wan), Hainan Provincial Foreign Experts Program (G20250218020E to Y.Wan), National Foreign Experts Individual Program (No. 110000212720258023 to Y.Wan), Hainan Province Science and Technology Special Fund (ZDYF2022XDNY246 to Y.Wan, ZDYF2025SHFZ061 to Y.Wan, ZDYF2024XDNY201 to F.S., ZDYF2023GXJS003 to Y.Wan and ZDYF2023XDNY074 to Z.Y.) and the Key Laboratory of quality safe evaluation and research of degradable material for State Market Regulation (No. 2022MK102 to Y.Wan). C.Mao would like to thank the Hong Kong Jockey Club Charities Trust for supporting the JC STEM Lab of Nature-Inspired Precision Medical Engineering (2022-0097). C.Mao also thanks the General Research Fund of Hong Kong (14208723), Collaborative Research Fund of Hong Kong (C4075-24G), and CUHK-CUHK(SZ)-GDSTC Joint Collaboration Fund (2025A0505000063) for financial support.

## Author contributions

B.W., Y.Wan, and C.Mao conceived the original idea. B.W., Y.Wan, X. Z., A. L., R.W., Y.Wei, F. S., Y.S., Z.Y., C.Ma, H. M., T.Y., H.H., Y.Z., and C.Mao planned and carried out the experiments. B.W., S.H., S.Z., R.W., X.Z., and Y.Wan designed and derived the mathematical model. S.L. offered serum samples and participated in the data discussion. B.W., Y.Wan, X. L., X. W., Q. W., R. L., D. Y., R. D., and C.Mao analyzed the data and took the lead in writing the manuscript. All authors provided critical feedback and helped shape the research, analysis, and manuscript.

## Competing interests

The authors declare no competing interests.
