## [Transparent Peer Review file · Nature Communications]

PRICE: direct and robust detection of microRNAs at single-nucleotide resolution

Corresponding Author: Professor Chuanbin Mao

Version 0:

Reviewer comments:

Reviewer #1

(Remarks to the Author)

The peptide nucleic acid (PNA)-mediated CRISPR/Cas13a system (PRICE) presented in this study offers a novel and innovative approach for miRNA at single-nucleotide resolution. The authors compare PRICE with artificially mismatched single-base CRISPR/Cas13a systems and qPCR for clinical sample detection. It sounds interesting. However, several aspects require further clarification before publication.

1. The clinical validation is limited to the let-7 miRNA family. The authors should clarify the choice of specific let-7 family members (let-7a, let-7c, let-7e, and let-7f) without including let-7b and let-7d. Additionally, it would be valuable to discuss the applicability of the PRICE system to other RNA types, such as snoRNAs and lncRNAs.
2. The authors mention the extension of the PRICE platform to Cas14f in the final discussion section, despite the term "PRICE" being derived from a PNA-mediated CRISPR/Cas13a system. This introduces a potential inconsistency, as it implies two separate platforms, given that the melting temperature (T_m) of PNA/RNA and PNA/DNA complexes differs. The authors should provide further clarification on how the PRICE system can be applied to Cas14f. Additionally, it is important to address whether the presence of DNA in clinical samples could affect the analytical performance of the system when extended to Cas14f.
3. The principle of the system is based on the melting temperature (T_m) difference, but the authors should clarify why the mutant/PNA complex exhibits higher T_m , while the miRNA/PNA complex shows a lower T_m . Furthermore, since Cas13a operates at around 37°C, it would be useful to explain how this temperature affects the performance of the system. Besides, the authors should provide a discussion of key reaction parameters (e.g., buffer composition, pH, salt concentration, and the use of blockers) in the context of SNV detection using the PRICE system. This would enhance the understanding of factors affecting performance.
4. The study investigates PNA length optimization within the 7–9 bp range, but it is unclear why this specific interval was chosen. The authors should justify their focus on lengths of 7, 8, and 9 bp, and consider expanding the investigation to include both shorter and longer PNA sequences to provide a more comprehensive understanding of the optimization process.
5. During clinical validation with serum samples (26 liver cancer-positive and 11 healthy), the number of positive samples is significantly higher than the number of negative samples. Typically, control sample sizes should be equal to or greater than positive sample sizes to ensure statistical robustness. The authors should address this imbalance in the sample sizes and provide a justification for their choice. Additionally, the statistical symbol "p" should be written as "P" throughout the manuscript.
6. Given that the PRICE system enhances sensitivity by blocking mutants, the authors should discuss the potential impact of various clinical sample interferences, including RNA, DNA, and other substances. Additionally, the article would benefit from a discussion of key performance metrics, such as detection time, sensitivity (limit of detection), and cost-effectiveness. This

would provide valuable insight into the system's practicality and feasibility for clinical use.

7. The article mentions reduced detection efficiency of PRICE when mutation sites are located at both ends of the miRNA. The authors should propose potential strategies or technical modifications to overcome this limitation.

8. Much of the content of the article is conveyed through figures, with limited textual description and interpretation, especially in the supplementary sections. Detailed captions and comprehensive discussions for each figure would improve clarity and data transparency.

Reviewer #2

(Remarks to the Author)

The authors formulated a new generalized CRISPR/Cas-based strategy, which they termed peptide nucleic acid (PNA) mediated CRISPR/Cas13a system (PRICE), with the objective to detect single nucleotide variations (SNVs) of a miRNA sequence without sacrificing sensitivity. They designed, tested, and optimized a panel of PNA blockers complementary to non-target miRNAs but not to the target miRNA. They also showed that the PNA blockers selectively hybridize with non-target miRNAs in samples (serum, cells, or tissues), while remaining unable to bind the target miRNA. The unhybridized target miRNA was shown to bind to crRNA in the Cas13a/crRNA complex, to activate the Cas13a and cleave the RNA linker between a fluorescent reporter and quencher, generating a fluorescent signal that reports the presence of the target miRNA with a limit of detection (LOD) of ~10 fM.

The work was thoughtfully conceived and executed, and this reviewer recommends publication after revision. Below are comments and recommendations by this reviewer.

Comments:

1. The authors use HepG2 as a cancer cell line and the LO2 as a healthy cell line. HepG2 is a recognized cancer cell line but LO2 is believed by some to be also a cancer cell line due to contamination with HeLa cells (see Reference a). The observation that the data with the HepG2 line show an apparent underexpression for the let7 family compared to the LO2 line can't be taken as evidence that the LO2 line is a healthy line (see Figs. 5b, c). The use of a cell line which is generally accepted as healthy line would be preferable.
2. The authors report an LOD for the PRICE technology of 10 fM which suggests a Limit of quantitation in the range of 30-50 fM. Accurate data are required to suggest over- or under-expression of a miRNA disease biomarker compared to healthy, and not just to report the presence/absence of a specific miRNA. Not many miRNAs exhibit equal or higher than 30 fM levels in blood and therefore the PRICE platform may not be suitable for liquid biopsy testing which is currently preferred over tumor biopsy. For example, the concentration of let-7b in serum has been reported at 10 fM and the concentration of miR-21, which is considered quite abundant, at 8.7 fM (see Reference b).
3. The authors report LOD for let-7a (at 24 fM) and LOD for let-7e (at 10 fM). Since let-7a and let-7e differ by a single nucleotide, it is unclear to this reviewer how LODs which differ by a factor of 2.4 have been obtained and accepted. The reported 2.4-fold LOD difference for these two molecules indicates issues with reproducibility and/or accuracy in PRICE measurements. As pointed out earlier, because the measurement needs to show over- or underexpression, accuracy and reproducibility of the analytical tool is of paramount importance.
4. Most of the reported data to show efficient hybridization between non-target miRNA and PNA-blocker were tested at pM and nM concentrations which may not be transferrable at the fM concentrations required in Diagnostics.
5. Synthesizing PNA-blockers for every miRNA to be measured and adding them at pM concentrations in a real sample is a rather complex and expensive proposition and may prohibit further exploration.
6. Despite the above-mentioned shortcomings of this study, this reviewer finds that the data are supportive in exploiting the PNA blockers to eliminate undesirable miRNA mutants, but this reviewer finds the use of either qPCR or CRISPR/Cas13a as suboptimal due to the observed large spread of the data (see comment No7 below). This reviewer proposes publication after revision.
7. The data reported by these authors (see figures 5d, e, k, l) exhibit large data spread within a certain cohort. Even though this is not atypical in the miRNA field, it creates uncertainty and has prohibited regulatory approvals for miRNA-based diagnostics and therapeutics. Recently a new analytical platform has surfaced which measures miRNAs directly using a nanopore-array for detection and a probe complementary to the target (see References b and c). Publications using this new platform illustrate the absence of sample variability and show comparable miRNA copy numbers from subjects of different sex, age, and ethnicity after copies are normalized to the same total RNA content in the sample. This analytical platform further claims that comparable miRNA copy number can be obtained from blood or urine. An approximate 2-fold overexpression of several known miRNA cancer biomarkers was observed in stage I/II serum samples from untreated patients upon cancer diagnosis. Zero data overlap between cancer and healthy samples was observed with a P-value 1.6×10^{-22} . This novel platform has raised many questions regarding the suitability of the of the current analytical tools, including the use of the analytical tools used in this study. This reviewer recommends that the authors normalize their data (see figures 5d, e, k, l) to the same total RNA content, where available, and include the revised data in this manuscript for comparison.

References cited above:

- (a) V Shao, T., & Chen, Y. L. (2024). Stop using the misidentified cell line LO2 as a human hepatocyte. *Journal of hepatology*, 80(5), e200–e201. <https://doi.org/10.1016/j.jhep.2023.10.036> and Song, Q., Zhang, X., & Yu, J. (2024). Reply to: "Stop using the misidentified cell line LO2 as a human hepatocyte". *Journal of hepatology*, 80(5), e202–e203. <https://doi.org/10.1016/j.jhep.2023.12.026>.

(b) Kanavarioti, A., Rehman, M. H., Qureshi, S., Rafiq, A., & Sultan, M. (2024). High Sensitivity and Specificity Platform to Validate MicroRNA Biomarkers in Cancer and Human Diseases. *Non-coding RNA*, 10(4), 42.

<https://doi.org/10.3390/ncrna10040042>.

(c) Rafiq, A., & Kanavarioti, A. (2025). The Potential and Limitations of the MinION/Yenos Platform for miRNA-Enabled Early Cancer Detection. *International journal of molecular sciences*, 26(8), 3822. <https://doi.org/10.3390/ijms26083822>

Version 1:

Reviewer comments:

Reviewer #1

(Remarks to the Author)

No further comments. The authors have satisfactorily addressed all previous questions. I agree that the manuscript is now suitable for publication.

Reviewer #2

(Remarks to the Author)

The authors formulated a new generalized CRISPR/Cas-based strategy, which they termed peptide nucleic acid (PNA) mediated CRISPR/Cas13a system (PRICE), with the objective to detect single nucleotide variations (SNVs) of a miRNA sequence without sacrificing sensitivity. They designed, tested, and optimized a panel of PNA blockers complementary to non-target miRNAs but not to the target miRNA. They also showed that the PNA blockers selectively hybridize with non-target miRNAs in samples (serum, cells, or tissues), while remaining unable to bind the target miRNA. The unhybridized target miRNA was shown to bind to crRNA in the Cas13a/crRNA complex, to activate the Cas13a and cleave the RNA linker between a fluorescent reporter and quencher, generating a fluorescent signal that reports the presence of the target miRNA with a limit of detection (LOD) of ~10 fM.

The work was thoughtfully conceived and executed. It represents a major improvement over current miRNA quantification technologies, and it will positively impact diagnostic efforts based on miRNA profiling. The authors responded clearly and by doing additional experiments to the comments made by this reviewer. This reviewer recommends publication without revision.

Point-to-point responses to comments of reviewers and editors:

REVIEWER COMMENTS

Reviewer #1 (Remarks to the Author):

The peptide nucleic acid (PNA)-mediated CRISPR/Cas13a system (PRICE) presented in this study offers a novel and innovative approach for miRNA at single-nucleotide resolution. The authors compare PRICE with artificially mismatched single-base CRISPR/Cas13a systems and qPCR for clinical sample detection. It sounds interesting. However, several aspects require further clarification before publication.

Reply:

We are very grateful to Reviewer's feedback.

1. The clinical validation is limited to the let-7 miRNA family. The authors should clarify the choice of specific let-7 family members (let-7a, let-7c, let-7e, and let-7f) without including let-7b and let-7d. Additionally, it would be valuable to discuss the applicability of the PRICE system to other RNA types, such as snoRNAs and lncRNAs.

Reply: We sincerely thank the reviewer for their insightful comments and valuable suggestions, which have helped us improve the manuscript. We address each point below:

We appreciate the reviewer's question regarding our selection of specific let-7 family members for clinical validation. Our selection was primarily based on two key factors: (1) Biological Significance: The let-7 family plays a pivotal role in post-transcriptional gene expression regulation, controlling many determination genes to influence cell decisions (Nam, Y., et al. Cell 2011, 147(5), 1080-1091; Zisoulis, D. G., et al. Nature 2012, 486(7404), 541-544; Wells, A. C., et al., Nat. Commun. 2023, 14, 5585)¹⁻³. Even minor variations or deregulation in these miRNAs can lead to far-reaching phenotypic and clinical consequences, including various cancers (Smits, K.M. et al. Clinical.Cancer.Research 2011, 17(24), 7723-7731; Li, Q., et al. Nat. Commun. 2017, 8(1), 1789)^{4,5}. (2) Technical Validation: High-quality, well-validated detection probes with proven specificity for these individual isoforms were available and optimized within our PRICE platform. Specifically, we selected let-7a, let-7c, let-

7e, and let-7f due to their single-nucleotide variations (SNVs) as shown in Fig. R1, which allowed us to effectively demonstrate the specificity and efficiency of PRICE for SNV identification. In contrast, let-7b and let-7d contain multiple base variations, complicating interpretation in this initial clinical validation.

Applicability of PRICE to other RNA types (snoRNAs, lncRNAs): We thank the reviewer for highlighting the potential of applying the PRICE system beyond miRNAs. The core strength of PRICE is its modular and adaptable design, which can be extended to other RNA classes such as small nucleolar RNAs (snoRNAs) or long non-coding RNAs (lncRNAs). This extension is feasible through careful design of PNA blocking probes following two criteria: (1) the melting temperature of PNA/target (T_{m_mis}) should not exceed 22.0 °C, with a lower T_{m_mis} being preferable for PRICE, and (2) the melting temperature of PNA/mutant (T_m) should not be lower than 40.0 °C.

While the primary focus of this study was the development and clinical validation of PRICE for specific miRNA targets (the let-7 family), we fully agree that its application to other functionally significant RNA classes represents a highly promising future direction. We have now added a paragraph to the Discussion section (page 18, second paragraph, marked in red in the updated manuscript) explicitly discussing this broader applicability:

"The modular architecture of the PRICE system suggests significant potential for adaptation beyond miRNAs. Future work will explore its application to other clinically relevant RNA classes, such as snoRNAs and lncRNAs. Key considerations for this adaptation include the design of target-specific capture probes optimized for potentially lower abundance or structurally complex targets, and potential adjustments to sample processing protocols. Successful extension to these RNA types could significantly broaden the diagnostic and research utility of the platform."

Let-7a	5'-UGAGGUAGUAGGUUGUAUAGUU
Let-7c	5'-UGAGGUAGUAGGUUGUAUGGUU
Let-7e	5'-UGAGGUAGGAGGUUGUAUAGU
Let-7f	5'-UGAGGUAGUAGAUUGUAUAGUU
Let-7b	5'-UGAGGUAGUAGGUUGUGUGGUU
Let-7d	5'-AGAGGUAGUAGGUUGCAUAGU

Fig. R1| Sequences of let-7 family members. The base variation sites of the sequences indicated in gray.

2. The authors mention the extension of the PRICE platform to Cas14f in the final discussion section, despite the term "PRICE" being derived from a PNA-mediated

CRISPR/Cas13a system. This introduces a potential inconsistency, as it implies two separate platforms, given that the melting temperature (T_m) of PNA/RNA and PNA/DNA complexes differs. The authors should provide further clarification on how the PRICE system can be applied to Cas14f. Additionally, it is important to address whether the presence of DNA in clinical samples could affect the analytical performance of the system when extended to Cas14f.

Reply: Thank you for your comments. The following summarizes the experimental procedure for the Cas14a platform:

We want to clarify that we did not state that our PRICE platform can be extended to Cas14a. In the paragraph where Cas14a was mentioned (page 16, paragraph 2), our purpose was to indicate that the use of PNA could improve the specificity of detecting single-base mutation in DNA in the traditional Cas14a platform. Indeed, our PRICE platform cannot be extended to the Cas14a platform. This is because we found that the detection sensitivity will be low (nM level) when PNA was integrated into the Cas14a platform. However, our PRICE system could reach very low sensitivity (fM level).

3. The principle of the system is based on the melting temperature (T_m) difference, but the authors should clarify why the mutant/PNA complex exhibits higher T_m , while the miRNA/PNA complex shows a lower T_m . Furthermore, since Cas13a operates at around 37°C, it would be useful to explain how this temperature affects the performance of the system. Besides, the authors should provide a discussion of key reaction parameters (e.g., buffer composition, pH, salt concentration, and the use of blockers) in the context of SNV detection using the PRICE system. This would enhance the understanding of factors affecting performance.

Reply: Thank you for your comments.

Firstly, regarding why the mutant/PNA complex exhibits a higher T_m , whereas the miRNA/PNA complex shows a lower T_m : this is because the PNA was designed to be fully complementary to the mutant sequence, thereby eliminating the impact of PRICE from the mutant. In contrast, a base mismatch exists in the target miRNA/PNA complex, which lowers its T_m .

Secondly, we have investigated key reaction parameters affecting the performance of the PRICE system, including buffer composition, pH, salt concentration, and the use of blockers, as shown in the Supplementary Figs. 6–7. Following your suggestion, we

have added a paragraph discussing these key parameters in the updated manuscript (page 11, paragraph 2, marked in red).

The added paragraph reads as follows:

“We further optimized key parameters that could potentially impact the performance of PRICE, including buffer type, pH, and salt concentration (Supplementary Figs. 6–7). The results showed that the choice of buffer (Tris-HCl or HEPES) had little effect on performance. However, an optimal pH of around 8.5 and a low salt concentration between 0 and 50 mM were preferred for best results.”

4. The study investigates PNA length optimization within the 7–9 bp range, but it is unclear why this specific interval was chosen. The authors should justify their focus on lengths of 7, 8, and 9 bp, and consider expanding the investigation to include both shorter and longer PNA sequences to provide a more comprehensive understanding of the optimization process.

Reply: Thank you for your comments. We investigated the impact of using longer PNAs (11nt and 13nt) on the performance of PRICE, testing them against both target and mutant sequences at a fixed molar ratio of 1:1. As shown in Fig. R2, while these longer PNAs effectively inhibited the signal from the non-target mutant miRNA, they also caused significant inhibition of the target signal—approximately 54% inhibition for the 11nt PNA, with the 13nt PNA causing even greater inhibition. Based on these results, we chose to use shorter PNAs (7–9nt) for optimizing PRICE for miRNA sensing. These data provide a strong basis for determining the optimal PNA design.

Regarding even shorter PNAs, such as 6nt, the calculated melting temperatures (using [https://www.pnabio.com/support/PNA\(Tool\).htm](https://www.pnabio.com/support/PNA(Tool).htm)) for PNAs binding to the mutant sequence were all below 37°C (Table. R1), which is the operating temperature for PRICE. This suggests that 6nt PNAs do not hybridize efficiently enough with the mutant to effectively eliminate its interference. Therefore, we did not include results for 6nt PNAs in our previous manuscript.

We incorporated these data into the updated supplementary file to follow your suggestions.

Fig. R2| Optimization of the PRICE assay for sensing miRNAs. (A-B), The impact of length of PNAs (13nt and 11nt) with target (T) or mutant (M) for the performance of PRICE. (B), kinetics measurements, (A), 1h end point measurements of (B). The molar ratio of target or mutant (10 pM) to PNA was 1:1. Data represent the mean \pm s.d., three technical replicates.

Table. R1| The melting temperatures of different PNAs (nt=6) with the mutant(T₁₀(A→G)).

NO#	Length (nt)	PNA sequence (N to C)	T _m of PNA/mutant (°C)
1	6	CAACCC	31.2
2	6	AACCCA	34.1
3	6	ACCCAC	33.4
4	6	CCCACT	27.9
5	6	CCACTA	24.3
6	6	CACTAC	23.2

Note: The red base indicates the location of the mutation.

5. During clinical validation with serum samples (26 liver cancer-positive and 11 healthy), the number of positive samples is significantly higher than the number of negative samples. Typically, control sample sizes should be equal to or greater than positive sample sizes to ensure statistical robustness. The authors should address this imbalance in the sample sizes and provide a justification for their choice. Additionally, the statistical symbol "p" should be written as "P" throughout the manuscript.

Reply: Thank you for your insightful comments on sample size balance and statistical notation. We address each point below:

We have gone through some of the most recent publications on nucleic acid detection in Nature journals and found that the number of positive samples can be greater than the negative samples (Table R2), just like the case in our study. Despite fewer controls, the differences between cancer-positive and healthy groups using the PRICE system were highly significant ($P < 0.001$, see Fig. 2K), indicating that the effect sizes for dysregulated let-7 miRNAs were substantial and appropriate for the cohort size, showing strong discriminatory power.

We appreciate the reviewer's note regarding the statistical notation. All instances of the symbol "p" have been corrected to uppercase "P" consistently throughout the manuscript.

Table. R2| The comparison of sample sizes between control and positive.

Applications	Control (healthy) sample sizes	Positive sample sizes	Ref
Single-cell immune profiling with COVID-19	5	13	(Zhang JY., et al. Nat Immunol 2020, 21, 1107–1118) ⁶
Diagnosis of bladder cancer	45	60	(Keum C., et al. Nat. Biomed. Eng 2025, 9, 1026–1038) ⁷
Detection of miRNAs in pancreatic ductal adenocarcinoma	15	20	(Yan H., et al. Nat. Biomed. Eng 2023, 7, 1583–1601) ⁸
miRNA detection of the bladder patient	5	10	(Moon J., Liu, C. Nat Commun 2023, 14, 7504) ⁹

6. Given that the PRICE system enhances sensitivity by blocking mutants, the authors should discuss the potential impact of various clinical sample interferents, including RNA, DNA, and other substances. Additionally, the article would benefit from a discussion of key performance metrics, such as detection time, sensitivity (limit of detection), and cost-effectiveness. This would provide valuable insight into the system's practicality and feasibility for clinical use.

Reply: Thank you for your valuable feedback. We appreciate your suggestions regarding clinical interferents and performance metrics, which will strengthen the practical relevance of our work.

The PRICE platform demonstrates robust resistance to interferents such as non-target RNA and DNA due to two primary features. First, the use of PNA probes, which possess an uncharged backbone, eliminates electrostatic repulsion and enables stronger and more specific hybridization to the target nucleic acids compared to

traditional DNA probes. This specificity is highlighted by PNA's ability to discriminate single-base mismatches, resulting in a significant melting temperature drop (8~20°C) for single mismatches and complete failure to hybridize when two mismatches occur. Moreover, the binding stability of PNA is maintained across various salt concentrations and is resistant to enzymatic degradation (Nandhini K.P., et al. *Chemical.Society.Reviews* 2023, 52, 2764-2789; Egholm M., et al. *Nature* 1993, 365, 566-568; Xiang Z., et al. *Advanced.Science* 2024, 11(33), 2310225)¹⁰⁻¹². Second, the integration of Cas proteins, well-established for their precision in clinical sensing, enhances the platform's robustness against complex biological matrices (Yan H., et al. *Nat. Biomed. Eng.* 2023, 7, 1583–1601; Yu J., et al. *Nat. Chem. Biol.* 2025; Wu X., et al. *Nat. Biomed. Eng.* 2025)^{8, 13, 14}. We now explicitly address in Discussion (page 17, paragraph 3, marked in red) how these combined characteristics effectively reduce interference from non-target nucleic acids and other biomolecules present in clinical samples.

To experimentally validate this robustness against such interference, we challenged the PRICE assay with a vast excess of non-specific DNA and RNA (XNA). As shown in Fig. R3, even in the presence of XNA at a 10,000-fold (10^4) higher concentration than the target and mutant miRNAs, the target signal remains unaffected. Furthermore, the false-positive signal is effectively suppressed by the PNA blocker. These findings directly demonstrate the assay's robustness in complex environments.

Regarding the key performance metrics, we acknowledge that detection time, sensitivity, and cost-effectiveness are vital for clinical applications. We have compared various techniques with PRICE, as shown in Table R3. These metrics support the practicality of PRICE for clinical use. Additionally, we have included a dedicated section in the Discussion (page 17, paragraph 3, highlighted in red) that discusses these advantages and a cost-benefit analysis (Table R4). This information has also been updated in our supplementary file (Supplementary Table 7-8).

Fig. R3| Effect of XNA (RNA+DNA) interference on the specificity enhancement of PRICE. (A) Sequences used in the experiments. (B) The impact of XNA on PRICE performance, with (C) showing the corresponding dynamic measurements. The target (T) and mutant (M) concentrations were both 1 pM, while the RNA and DNA from the XNA mixture were both 10 nM, which is 10,000 times (10^4) higher than the target and mutant. Data represent the mean \pm s.d., $n=3$ technical replicates.

Table. R3| Comparison of PRICE with other exemplary detection techniques.

Technique	Detection time	Limit of detection	Cost per test	Ref
mCARMEN	5 h	~ 0.166 - 1.66 fM	\$13	(Welch N.L., et al. Nat Med 2022, 28, 1083–1094) ¹⁵
EXTRA-CRISPR	20 min - 3 h	12.3 - 1.64 fM	\geq \$0.60	(Yan H., et al. Nat. Biomed. Eng 2023, 7, 1583–1601) ⁸
TEQUILA-seq	≥ 4 h	180 fM	\$0.24	(Wang F., et al. Nat Commun 2023, 14, 4760) ¹⁶
Cas12a-enhanced RCA	4.5 h	34.7 fM	\$3.70	(Zhang G., et al. Microchemical Journal 2020, 158) ¹⁷
Cas12a-mediated cascade amplification	~ 4 h	21.9 fM	\$3.50	(Sun H., et al. Analyst 2020, 145, 5547-5552) ¹⁸
PRICE	~ 3.5 h	~ 10 fM	\$5.32	This work

Table. R4| Main cost of the PRICE.

	Amount	PRICE (RMB)	Concentration in 20uL	Cost per reaction (RMB)	Cost per reaction (Dollar)
RNA extraction kit	50 Times	1850	\	37	5.153
PNA	20 nMol	950	1 nM	$9.5 \times (10^{-4})$	\

Cas13a	500 pMol	2500	20 nM	1	0.139
crRNA	3.7 nMol	520	20 nM	0.056	0.008
Probe	25.6 nMol	700	300 nM	0.164	0.023
Note: The price was calculated based on the exchange rate of 1 USD to 7.18 RMB as of August 8, 2025.				Total: 38.22	Total: 5.324

7. The article mentions reduced detection efficiency of PRICE when mutation sites are located at both ends of the miRNA. The authors should propose potential strategies or technical modifications to overcome this limitation.

Reply: We sincerely thank the reviewer for highlighting the important challenge of detecting mutations located at the terminal ends of miRNAs using PRICE. This is a recognized limitation of CRISPR-based detection systems, and we propose several targeted strategies, informed by recent advances, that could significantly enhance PRICE's detection capability for terminal mutations:

(1) Optimized guide RNA design: By leveraging machine learning and generative algorithms such as BADGERS, artificial guide RNAs can be strategically engineered to position mismatches adjacent to the target single-nucleotide polymorphisms. This approach improves target specificity by refining the binding dynamics of the guide RNA (Mantena S., et al. *Nat.Biotechnol.*2024, 1-8)¹⁹.

(2) Cas enzyme engineering: Through amino acid substitutions, the mismatch tolerance of Cas enzymes can be fine-tuned to increase specificity for terminal mutations. Directed evolution techniques, including phage-assisted continuous evolution, have successfully enhanced Cas enzyme specificity, thereby improving cleavage accuracy at challenging sites (Duan Z., et al. *The.Innovation* 2024, 5(2); Liu Y., et al. *Nat.Commun* 2024, 15, 5014; Hu J., et al. *Nature* 2018, 556, 57–63)²⁰⁻²².

(3) PNA block modification: Chemical modifications to the PNA backbone, such as incorporating lysine, alanine, or glutamic acid at the γ -carbon position, can enhance binding affinity and stability. This improved binding efficiency can compensate for the inherently weaker interactions near terminal mutation sites.

Importantly, a combined implementation of these strategies holds promising potential to overcome the current detection limitations of PRICE. We will explore these modifications in future optimizations to further improve assay sensitivity and specificity.

We have incorporated potential strategies into the updated manuscript (page 16, paragraph 1, highlighted in red) as follows:

“To overcome this limitation, several complementary strategies can be employed: (7) optimizing guide RNA design using machine learning and generative algorithms like BADGERS; (8) engineering the Cas enzyme through amino acid substitutions or directed evolution methods including phage-assisted continuous evolution and (9) chemically modifying the PNA backbone for example by introducing lysine, alanine, or glutamic acid at the i₁ carbon position. Integrating these approaches offers a promising path to enhance PRICE's performance and address its current constraints.”

8. Much of the content of the article is conveyed through figures, with limited textual description and interpretation, especially in the supplementary sections. Detailed captions and comprehensive discussions for each figure would improve clarity and data transparency.

Reply: Thank you for the suggestion. We appreciate the emphasis on clarity and transparency. We have revised the text as follows:

(1) We expanded all figure captions to explicitly state the key experimental conditions, main observations, and the conclusions supported by each figure.

(2) We added dedicated interpretive text in both the main Results/Discussion sections and in the Supplementary Text for each figure. These additions explain the figure's purpose, the specific insights it provides, any limitations, and how it supports the study's overall conclusions.

All new or revised text is highlighted in red in the updated file for easy review.

Reviewer #2 (Remarks to the Author):

The authors formulated a new generalized CRISPR/Cas-based strategy, which they termed peptide nucleic acid (PNA) mediated CRISPR/Cas13a system (PRICE), with

the objective to detect single nucleotide variations (SNVs) of a miRNA sequence without sacrificing sensitivity. They designed, tested, and optimized a panel of PNA blockers complementary to non-target miRNAs but not to the target miRNA. They also showed that the PNA blockers selectively hybridize with non-target miRNAs in samples (serum, cells, or tissues), while remaining unable to bind the target miRNA. The unhybridized target miRNA was shown to bind to crRNA in the Cas13a/crRNA complex, to activate the Cas13a and cleave the RNA linker between a fluorescent reporter and quencher, generating a fluorescent signal that reports the presence of the target miRNA with a limit of detection (LOD) of ~10 fM. The work was thoughtfully conceived and executed, and this reviewer recommends publication after revision. Below are comments and recommendations by this reviewer.

Reply:

We are very grateful to Reviewer's feedback.

Comments:

1. The authors use HepG2 as a cancer cell line and the LO2 as a healthy cell line. HepG2 is a recognized cancer cell line but LO2 is believed by some to be also a cancer cell line due to contamination with HeLa cells (see Reference a). The observation that the data with the HepG2 line show an apparent underexpression for the let7 family compared to the LO2 line can't be taken as evidence that the LO2 line is a healthy line (see Figs. 5b, c). The use of a cell line which is generally accepted as healthy line would be preferable.

Reply: We sincerely thank the reviewer for identifying the critical concern regarding LO2's potential HeLa contamination, which undermines its validity as a healthy control; to rectify this, we have replaced LO2 with the well-characterized normal human hepatocyte line THLE-2 (ATCC, CRL-2706)—a widely accepted non-malignant model (Bae, H., et.al. Oncogene 2014, 33, 2557–2567; Fernández-Tussy P., et.al. Cell. Death.Dis 2021, 12, 555; Huang X., et.al; Nat.Commun 2023, 14, 4620)²³⁻²⁵—repeated all comparative experiments with HepG2, and confirmed that the let-7 family remains significantly under expressed in cancer cells (updated Fig.R4), thereby preserving our original conclusion while strengthening the biological validity of our findings through the use of an uncontaminated, generally accepted healthy cell line, with all figures, results, and methodological descriptions updated in the manuscript.

Fig. R4| PRICE analysis for the miRNA regulation in cell cultures. (A-B), The expression levels of Hsa-let-7e (A) and Hsa-let-7f (B) in THLE-2 and HepG2 cell lines were measured by PRICE and qPCR methods, respectively (paired two-tailed Student's t test, $**P < 0.01$, $****P < 0.0001$). (C-D) Kinetic measurements for the miRNA regulation in cell culture. Data corresponding to (A) and (B), respectively. HepG2 cell was the cancer positive sample, while THLE-2 cell was used as control negative to illustrate the miRNA regulation. The control represents the signal in the absence of the target miRNA. Data represent the mean \pm s.d., $n=3$ technical replicates.

2. The authors report an LOD for the PRICE technology of 10 fM which suggests a Limit of quantitation in the range of 30-50 fM. Accurate data are required to suggest over- or under-expression of a miRNA disease biomarker compared to healthy, and not just to report the presence/absence of a specific miRNA. Not many miRNAs exhibit equal or higher than 30 fM levels in blood and therefore the PRICE platform may not be suitable for liquid biopsy testing which is currently preferred over tumor biopsy. For example, the concentration of let-7b in serum has been reported at 10 fM and the concentration of miR-21, which is considered quite abundant, at 8.7 fM (see Reference b).

Reply: We appreciate the reviewer's valid concern regarding miRNA abundance in serum and confirm that PRICE's current 10 fM LOD is indeed sufficient for detecting let-7 family miRNAs in blood samples. This capability is achieved through our target enrichment strategy: By using magnetic nanoparticles functionalized with target-capturing DNA, we can magnetically concentrate the target miRNAs from 1 mL serum into 50 μ L, effectively increasing their concentration *20-fold* (e.g., a serum concentration of \sim 10 fM let-7b becomes \sim 200 fM in the assayed sample, and 8.7 fM miR-21 becomes 174 fM—both well above our LOD, which is \sim 10 fM). This approach, validated by studies using comparable LOD methods for serum detection, as showed in Table. R5 (Cai S., et al. *Nat. Commun* 2021,12, 3515; Broto M., et al. *Nat. Nanotechnol*; 2022, 17, 1120–1126; Chen Y., et al. *Nat. Commun* 2024,15, 8342; Wang J., et al. *Nat. Commun* 2025,16, 6392)²⁶⁻²⁹, ensures robust quantification of biologically relevant expression differences between healthy and diseased states. While we acknowledge that extending PRICE to lower-abundance miRNAs would require further sensitivity enhancements (as noted in our Discussion (page 17, paragraph 2): Cas engineering, signal amplification, or advanced enrichment), the current platform remains rigorously validated for its stated targets in liquid biopsy applications.

Table. R5| LOD comparison of different sensing methods.

Target	LOD (fM)	Ref
miR-141/miR-375	5 / 8	(Cai S., et al. Nat Commun 2021,12, 3515) ²⁶
Inc-LIPCAR	8.3	(Broto M., et al. Nat. Nanotechnol. 2022, 17, 1120–1126) ²⁷
RNA/DNA	\sim 100	(Chen Y., et al. Nat Commun 2024,15, 8342) ²⁸
miR-21/miR-17/miR-31/miR-92a	26.7 /37.2 /33.5 /28.9	(Wang J., et al. Nat Commun 2025,16, 6392) ²⁹
Let-7	\sim 10	This work

3. The authors report LOD for let-7a (at 24 fM) and LOD for let-7e (at 10 fM). Since let-7a and let-7e differ by a single nucleotide, it is unclear to this reviewer how LODs which differ by a factor of 2.4 have been obtained and accepted. The reported 2.4-fold LOD difference for these two molecules indicates issues with reproducibility and/or accuracy in PRICE measurements. As pointed out earlier, because the

measurement needs to show over- or underexpression, accuracy and reproducibility of the analytical tool is of paramount importance.

Reply: Thank you for your comment. We appreciate the opportunity to clarify the observed difference in the LODs between let-7a and let-7e.

It is important to note that the sensitivity of CRISPR-based detection systems (for example, SHERLOCK) depends not only on target sequence identity but also on contextual factors such as guide–target mismatches, local secondary structure, and sequence-dependent cleavage kinetics. Even a single-nucleotide difference can markedly change Cas enzyme binding, cleavage efficiency, and signal output because the position and type of a mismatch affect Cas12a/Cas13a activity, RNA/DNA folding can occlude or expose target sites, and different sequences elicit varying rates of collateral cleavage activation. Consequently, LODs can vary substantially between targets; for example, LODs on the SHERLOCK platform have been reported to differ by more than 40-fold across target sequences (see red frame in Fig. R5, De Puig et al., *Sci. Adv.* 2021)³⁰. Similarly, using the same gRNA to detect two strains that differ by two bases, or using two gRNAs that differ by two bases to detect the same target, can change signal output by one to over two orders of magnitude (see Fig. R6, Gootenberg et al., *Science* 2017)³¹.

In our experiments, the 2.4-fold difference observed between let-7a and let-7e is within the expected range of variation for Cas-based assays and therefore does not indicate poor reproducibility or inaccuracy. Moreover, to further validate the reproducibility and accuracy, we repeated the LOD measurements for let-7a and let-7e using our platform. As shown in Fig. R7, the results were consistent with those in Figs. 3e-f of our manuscript, demonstrating high sensitivity and specificity for miRNA detection and underscoring the robustness of the PRICE platform.

We agree that accuracy and reproducibility are critical for diagnostic applications, and we have taken great care to optimize and validate each assay under controlled conditions. The observed LOD variations are a known characteristic of Cas enzyme behavior and do not detract from the clinical utility of the platform.

[Figure Redacted]

Fig. R5| Performance of the miSHERLOCK device for SARS-CoV-2. (A) SARS-CoV-2 spike genomic map indicating the target sequences and selected gRNA sequences used in this study. (B) Table summarizing performance near the LOD of four SARS-CoV-2 targets tested in the miSHERLOCK device; the red frame marks the LOD for each target sequence. (De Puig H, et al. *Sci. Adv.* 2021, 7, 2944)³⁰

[Figure Redacted]

Fig. R6| Cas13a detection between similar viral strains. (A, C) Schematic of ZIKV strain (or DENV strain) target regions and the crRNA sequences used for detection, respectively. SNPs in the target are highlighted red or blue, and synthetic mismatches in the guide sequence are in red. (B, D) Highly specific detection of strain SNPs allows for the differentiation of ZIKV African versus American RNA (DENV strain 1 versus strain 3 RNA) targets with Cas13a. (Gootenberg, et al. Science 2017,356, 438–442)³¹

Fig. R7| Reproducibility and accuracy of PRICE for measuring LOD. Sensing target of let-7a (A,C) and let-7e (B,D) at different concentrations using PRICE. (A-B) show column data, while (C-D) show the corresponding dynamic measurements. The concentration of PNA used was 1000-fold higher than that of the sensing targets (paired two-tailed Student's t test, $P < 0.05$). The control represents the background signal in the absence of the target miRNA. Data represent the mean \pm s.d., $n=3$ technical replicates.

4. Most of the reported data to show efficient hybridization between non-target

miRNA and PNA-blocker were tested at pM and nM concentrations which may not be transferrable at the fM concentrations required in Diagnostics.

Reply: We sincerely thank the reviewer for raising this important point regarding the potential concentration-dependent efficiency of PNA-blocker hybridization.

As we explained in the response to your Q2, we used magnetic nanoparticles to capture and enrich the target miRNAs before we apply our PRICE method for the detection of clinical samples. Specifically, by using magnetic nanoparticles functionalized with target-capturing DNA, we can magnetically concentrate the target miRNAs from 1 mL serum into 50 μ L, effectively increasing their concentration *20-fold* (e.g., a serum concentration of \sim 10 fM let-7b becomes \sim 200 fM in the assayed sample, and 8.7 fM miR-21 becomes 174 fM—both well above our LOD, which is \sim 10 fM). Namely, by magnetic enrichment, the otherwise less abundant clinical samples become more concentrated to reach a level detectable by our PRICE method. Therefore, our PRICE method can be potentially translated to clinical diagnostics.

5. Synthesizing PNA-blockers for every miRNA to be measured and adding them at pM concentrations in a real sample is a rather complex and expensive proposition and may prohibit further exploration.

Reply: We sincerely thank the reviewer for raising this important practical consideration regarding the complexity and cost associated with synthesizing PNA-blockers for each mutant miRNA. We acknowledge that the incorporation of sequence-specific PNA-blockers adds an additional step to the assay setup.

However, this strategy was intentionally adopted to tackle the daunting challenge in specifically discriminating single nucleotide variations (SNVs) without compromising sensitivity. While the initial design and synthesis of PNAs require investment, the resulting gains in the specificity of SNV detection significantly reduce the risk of false positives, thereby improving diagnostic reliability.

It is also worth noting that once designed and validated, these PNA-blockers can be produced at scale and used reproducibly across many clinical samples, thereby amortizing the initial development cost, as illustrated in Table. R6-7.

We agree that assay simplicity is always desirable, but in cases where extreme specificity is essential, for example, to distinguish miRNA isoforms that differ by only a single nucleotide regardless of its position, the use of PNA blockers provides a reliable and justifiable solution. We believe this approach achieves an optimal

balance between analytical performance and practical feasibility for precision diagnostics.

Table. R6| Comparison of PRICE with other exemplary detection techniques.

Technique	Detection time	Limit of detection	Cost per test	Ref
mCARMEN	5 h	~0.166-1.66 fM	\$13	(Welch N.L., et al. Nat Med 2022, 28, 1083–1094) ¹⁵
EXTRA-CRISPR	20 min - 3 h	12.3 -1.64 fM	≥ \$0.60	(Yan H., et al. Nat. Biomed. Eng 2023, 7, 1583–1601) ⁸
TEQUILA-seq	≥4 h	180 fM	\$0.24	(Wang F., et al. Nat Commun 2023, 14, 4760) ¹⁶
Cas12a-enhanced RCA	4.5 h	34.7 fM	\$3.70	(Zhang G., et al. Microchemical Journal 2020, 158) ¹⁷
Cas12a-mediated cascade amplification	~ 4 h	21.9 fM	\$3.50	(Sun H., et al. Analyst 2020, 145, 5547-5552) ¹⁸
PRICE	~ 3.5 h	~ 10 fM	\$5.32	This work

Table. R7| Main cost of the PRICE.

	Amount	PRICE (RMB)	Concentration in 20uL	Cost per reaction (RMB)	Cost per reaction (Dollar)
RNA extraction kit	50 Times	1850	\	37	5.153
PNA	20 nMol	950	1 nM	$9.5 \times (10^{-4})$	\
Cas13a	500 pMol	2500	20 nM	1	0.139
crRNA	3.7 nMol	520	20 nM	0.056	0.008
Probe	25.6 nMol	700	300 nM	0.164	0.023

Note: The price was calculated based on the exchange rate of 1 USD to 7.18 RMB as of August 8, 2025.

Total: 38.22

Total: 5.324

6. Despite the above-mentioned shortcomings of this study, this reviewer finds that the data are supportive in exploiting the PNA blockers to eliminate undesirable miRNA mutants, but this reviewer finds the use of either qPCR or CRISPR/Cas13a as suboptimal due to the observed large spread of the data (see comment No7 below). This reviewer proposes publication after revision.

Reply: We sincerely thank the reviewer for their constructive feedback and overall support of our approach using PNA-blockers to eliminate undesirable miRNA

mutants, as well as for their recommendation for publication after revision. Indeed, our PRICE method outperforms qPCR or CRISPR/Cas13a.

7. The data reported by these authors (see figures 5d, e, k, l) exhibit large data spread within a certain cohort. Even though this is not atypical in the miRNA field, it creates uncertainty and has prohibited regulatory approvals for miRNA-based diagnostics and therapeutics. Recently a new analytical platform has surfaced which measures miRNAs directly using a nanopore-array for detection and a probe complementary to the target (see References b and c). Publications using this new platform illustrate the absence of sample variability and show comparable miRNA copy numbers from subjects of different sex, age, and ethnicity after copies are normalized to the same total RNA content in the sample. This analytical platform further claims that comparable miRNA copy number can be obtained from blood or urine. An approximate 2-fold overexpression of several known miRNA cancer biomarkers was observed in stage I/II serum samples from untreated patients upon cancer diagnosis. Zero data overlap between cancer and healthy samples was observed with a P-value 1.6×10^{-22} . This novel platform has raised many questions regarding the suitability of the of the current analytical tools, including the use of the analytical tools used in this study. This reviewer recommends that the authors normalize their data (see figures 5d, e, k, l) to the same total RNA content, where available, and include the revised data in this manuscript for comparison.

References cited above:

- (a) V Shao, T., & Chen, Y. L. (2024). Stop using the misidentified cell line LO2 as a human hepatocyte. *Journal of hepatology*, 80(5), e200–e201. <https://doi.org/10.1016/j.jhep.2023.10.036> and Song, Q., Zhang, X., & Yu, J. (2024). Reply to: "Stop using the misidentified cell line LO2 as a human hepatocyte". *Journal of hepatology*, 80(5), e202–e203. <https://doi.org/10.1016/j.jhep.2023.12.026>.
- (b) Kanavarioti, A., Rehman, M. H., Qureshi, S., Rafiq, A., & Sultan, M. (2024). High Sensitivity and Specificity Platform to Validate MicroRNA Biomarkers in Cancer and Human Diseases. *Non-coding RNA*, 10(4), 42. <https://doi.org/10.3390/ncrna10040042>.
- (c) Rafiq, A., & Kanavarioti, A. (2025). The Potential and Limitations of the MinION/Yenos Platform for miRNA-Enabled Early Cancer Detection. *International journal of molecular sciences*, 26(8), 3822. <https://doi.org/10.3390/ijms26083822>

Reply: We sincerely thank the reviewer for their insightful comments and for highlighting the innovative nanopore-based miRNA detection platform, which demonstrates the value of normalization to total RNA content to minimize variability. We appreciate the opportunity to address this point and have cited the proper references suggested by this reviewer.

As mentioned above, in this study we standardized input material by using fixed sample quantities (approximately 70 mg per tissue sample and 1 mL per serum sample) to control for variations in sample processing. We acknowledge that, as is common in miRNA research, some degree of data overlap between cohorts can occur, as seen in Figures 5d, e, k, and l. This can be attributed to the fact that each data point represents one patient. Such phenomenon has also been observed in other studies, as illustrated in Fig. R8 (Ju CW., et al. *Nat. Biotechnol.* 2025; Keum C., et al. *Nat. Biomed. Eng.* 2025, 9, 1026–1038; Yan H., et al. *Nat. Biomed. Eng.* 2023, 7, 1583–1601)^{7, 8, 32}. While we recognize that normalization to total RNA content can serve as an additional robust strategy to reduce inter-sample variability—as powerfully demonstrated in the studies referenced by the reviewer—unfortunately, total RNA concentration measurements were not routinely performed for each individual sample in this specific dataset, making such normalization unfeasible at this stage.

However, each data point in Figures 5d, e, k, and l represents the mean of at least three technical replicates from a single patient. When all measurements are plotted, a clear downregulation of let-7e and let-7f in the liver patients can be observed (Fig. R9), validating our conclusion.

We fully agree that the approach you mentioned represents an important methodological refinement for future studies. To address the concern regarding data spread and to enhance the reliability of our findings, we are committed to incorporating total RNA quantification and normalization in all subsequent work. Meanwhile, we have cited the papers you mentioned in our manuscript to illustrate different data analyzing approach (page 15, paragraph 2, marked in red). In the revised manuscript, we added a detailed sample-processing procedure (page 22, paragraph 5, marked in red), addressed the impact of sample variability and normalization strategies, and placed our results in the context of current miRNA detection method (page 15, paragraph 2).

Thank you again for this constructive suggestion, which strengthens the framework of our ongoing research.

[Figure Redacted]

Fig. R8| Signal comparison of positive samples versus controls for each method. (A) LIME-seq detects RNA modification signatures in microbiome-derived RNAs in plasma cfRNA. Box plot comparing integral mutation patterns in microbiome-derived RNAs, showing significant differences between participants with colorectal cancer (CRC) and noncancer individuals across the three representative microorganisms in plasma cfRNA, validating the CRC diagnostic model (n = 36 for noncancer controls and n = 27 for CRC groups) (Ju CW., et al. *Nat.Biotechnol* 2025)³². (B) Evaluation of BLOOM for bladder cancer (BC) screening using clinical patient samples. NFIs of urine samples acquired from a BLOOM assay of patients with BC, patients with NMIBCs (n = 47 clinical urine samples) and patients with MIBCs (n = 13 clinical urine samples) versus no-tumour controls, including patients with genitourinary diseases (n = 20 clinical urine samples) and healthy controls (n = 25 clinical urine samples).

The lines represent the mean, and the error bars represent s.d. Two-tailed Student's t-tests were used for comparisons. ****P < 0.0001. Fluorescence intensities were normalized to that of the control (NPU) recorded without Hdase (I-6) (Keum C., et al. *Nat;Biomed;Eng* 2025,9, 1026–1038)⁷. (C) One-pot miRNA analysis for the diagnosis of pancreatic cancer. Scatter plots of individual sEV-miRNA markers and EV-Sig for discriminating the PDAC group from the control group. The expression levels of individual miRNAs in the isolated plasma EVs from the patients with PDAC (n = 20) and healthy donors (n = 15) measured by EXTRA-CRISPR. Each miRNA in each sample was tested in two technical replicates, and the background-subtracted signals were adjusted by that of the positive control. The middle line and error bar represent the mean and 1 s.e.m., respectively. P values were calculated by two-tailed Student's t-test with Welch correction (Yan H., et al. *Nat;Biomed;Eng* 2023,7, 1583–1601)⁸.

Fig. R9| PRICE analysis for the miRNA regulation in liver cancer tissue. The expression levels of Hsa-let-7e (A) and Hsa-let-7f (B) in liver tissue biopsies from patients (n=16) diagnosed with liver cancer were measured using the PRICE and qPCR techniques, respectively. The control samples were the peritumoral tissue located 1 centimeter away from the tumor mass, while the liver cancer samples were derived from the tumor mass tissue within the same patient. Each dot represents one measurement.

References cited in this response:

1. Nam, Y., Chen, C., Gregory, Richard I., Chou, James J. & Sliz, P. Molecular Basis for Interaction of let-7 MicroRNAs with Lin28. *Cell* 147, 1080-1091 (2011).
2. Zisoulis, D.G., Kai, Z.S., Chang, R.K. & Pasquinelli, A.E. Autoregulation of microRNA biogenesis by let-7 and Argonaute. *Nature* 486, 541-544 (2012).
3. Wells, A.C. et al. Let-7 enhances murine anti-tumor CD8 T cell responses by promoting memory and antagonizing terminal differentiation. *Nat. Commun.* 14, 5585 (2023).
4. Smits, K.M. et al. A Let-7 MicroRNA SNP in the KRAS 3'UTR Is Prognostic in Early-Stage Colorectal Cancer. *Clin. Cancer Res.* 17, 7723-7731 (2011).
5. Li, Q. et al. Cellular microRNA networks regulate host dependency of hepatitis C virus infection. *Nat. Commun.* 8, 1789 (2017).
6. Zhang, J.-Y. et al. Single-cell landscape of immunological responses in patients with COVID-19. *Nat. Immunol.* 21, 1107-1118 (2020).
7. Keum, C. et al. Diagnosis of early-stage bladder cancer via unprocessed urine samples at the point of care. *Nat. Biomed. Eng.* (2024).
8. Yan, H. et al. A one-pot isothermal Cas12-based assay for the sensitive detection of microRNAs. *Nat. Biomed. Eng.* 7, 1583-1601 (2023).
9. Moon, J. & Liu, C. Asymmetric CRISPR enabling cascade signal amplification for nucleic acid detection by competitive crRNA. *Nat. Commun.* 14, 7504 (2023).
10. Nandhini, K.P., Shaer, D.A., Albericio, F. & de la Torre, B.G. The challenge of peptide nucleic acid synthesis. *Chem. Soc. Rev.* 52, 2764-2789 (2023).
11. Egholm, M. et al. PNA hybridizes to complementary oligonucleotides obeying the Watson-Crick hydrogen-bonding rules. *Nature* 365, 566-568 (1993).
12. Xiang, Z. et al. Engineering of a DNA/γPNA Hybrid Nanoreporter for ctDNA Mutation Detection via γPNA Urinalysis. *Adv. Sci.* 11, 2310225 (2024).
13. Yu, J. et al. Programmable RNA acetylation with CRISPR-Cas13. *Nat. Chem. Biol.* (2025).
14. Wu, X. et al. LbuCas13a directly targets DNA and elicits strong trans-cleavage activity. *Nat. Biomed. Eng.* (2025).

15. Welch, N.L. et al. Multiplexed CRISPR-based microfluidic platform for clinical testing of respiratory viruses and identification of SARS-CoV-2 variants. *Natj.Medj*; 28, 1083-1094 (2022).
16. Wang, F. et al. TEQUILA-seq: a versatile and low-cost method for targeted long-read RNA sequencing. *Natj.Communj*; 14, 4760 (2023).
17. Zhang, G., Zhang, L., Tong, J., Zhao, X. & Ren, J. CRISPR-Cas12a enhanced rolling circle amplification method for ultrasensitive miRNA detection. *MicrochemjJj*; 158, 105239 (2020).
18. Sun, H.-H., He, F., Wang, T., Yin, B.-C. & Ye, B.-C. A Cas12a-mediated cascade amplification method for microRNA detection. *Analyst* 145, 5547-5552 (2020).
19. Mantena, S. et al. Model-directed generation of artificial CRISPR-Cas13a guide RNA sequences improves nucleic acid detection. *Natj.Biotechnolj*; (2024).
20. Duan, Z. et al. An engineered Cas12i nuclease that is an efficient genome editing tool in animals and plants. *The.Innovation* 5, 100564 (2024).
21. Liu, Y. et al. CoHIT: a one-pot ultrasensitive ERA-CRISPR system for detecting multiple same-site indels. *Natj.Communj*; 15, 5014 (2024).
22. Hu, J.H. et al. Evolved Cas9 variants with broad PAM compatibility and high DNA specificity. *Nature* 556, 57-63 (2018).
23. Fernández-Tussy, P. et al. Anti-miR-518d-5p overcomes liver tumor cell death resistance through mitochondrial activity. *Cell.Death.Disj*; 12, 555 (2021).
24. Bae, H.J. et al. MicroRNA-29c functions as a tumor suppressor by direct targeting oncogenic SIRT1 in hepatocellular carcinoma. *Oncogene* 33, 2557-2567 (2014).
25. Huang, X. et al. Cullin-associated and neddylation-dissociated protein 1 (CAND1) alleviates NAFLD by reducing ubiquitinated degradation of ACAA2. *Natj.Communj*; 14, 4620 (2023).
26. Cai, S. et al. Single-molecule amplification-free multiplexed detection of circulating microRNA cancer biomarkers from serum. *Natj.Communj*; 12, 3515 (2021).
27. Broto, M. et al. Nanozyme-catalysed CRISPR assay for preamplification-free detection of non-coding RNAs. *Natj.Nanotechnolj*; (2022).
28. Chen, Y. et al. Split crRNA with CRISPR-Cas12a enabling highly sensitive and multiplexed detection of RNA and DNA. *Natj.Communj*; 15, 8342 (2024).
29. Wang, J. et al. Regulating cleavage activity and enabling microRNA detection with split sgRNA in Cas12b. *Natj.Communj*; 16, 6392 (2025).
30. de Puig, H. et al. Minimally instrumented SHERLOCK (miSHERLOCK) for CRISPR-based point-of-care diagnosis of SARS-CoV-2 and emerging variants. *SciJ.Advj* 7, eabh2944 (2021).
31. Gootenberg, J.S. et al. Nucleic acid detection with CRISPR-Cas13a/C2c2. *Science* 356, 438-442 (2017).
32. Ju, C.-W. et al. Modifications of microbiome-derived cell-free RNA in plasma discriminates colorectal cancer samples. *Natj.Biotechnolj*; (2025).